# ACTL6A protects gastric cancer cells against ferroptosis through induction of glutathione synthesis

Ziqing Yang[1,2,3,4], Shaomin Zou[1,2,3,4], Yijing Zhang[1,2,3], Jieping Zhang[1,2,3], Peng Zhang[1,2,3], Lishi Xiao[1,2,3], Yunling Xie [1,2,3], Manqi Meng[1,2,3], Junyan Feng[1,2,3], Liang Kang [1,2] ✉, Mong-Hong Lee [1,2,3] ✉ & Lekun Fang [1,2,3] ✉

Gastric cancer (GC), one of the most common malignant tumors in the world, exhibits a rapid metastasis rate and causes high mortality. Diagnostic markers and potential therapeutic targets for GCs are urgently needed. Here we show that Actin-like protein 6 A (ACTL6A), encoding an SWI/SNF subunit, is highly expressed in GCs. ACTL6A is found to be critical for regulating the glutathione (GSH) metabolism pathway because it upregulates γ-glutamyl-cysteine ligase catalytic subunit (GCLC) expression, thereby reducing reactive oxygen species (ROS) levels and inhibiting ferroptosis, a regulated form of cell death driven by the accumulation of lipid-based ROS. Mechanistic studies show that ACTL6A upregulates GCLC as a cotranscription factor with Nuclear factor (erythroid-derived 2)-like 2 (NRF2) and that the hydrophobic region of ACTL6A plays an important role. Our data highlight the oncogenic role of ACTL6A in GCs and indicate that inhibition of ACTL6A or GCLC could be a potential treatment strategy for GCs.

Gastric cancer (GC) ranks as the fifth most frequently diagnosed cancer and the fourth most common cause of cancer death globally[1]. Recently, analysis of high-throughput and genomic technology allowed GCs to be studied at the molecular level. Molecular profiling data have greatly facilitated the identification of candidate gene driver alterations in GC, such as gene mutations, chromosomal alterations, transcriptional changes, and dysregulated epigenetic modifications[2,3]. Studies have shown that compared with more traditional histopathological classifications, cancer classifications based on molecular data may prove to be more clinically impactful in terms of predicting treatment and patient prognosis[4,5]. Therefore, the identification of more GC biomarkers may increase the survival time of GC patients.

Actin-like protein 6 A (ACTL6A), also known as 53 kDa BRG-1/ human BRM-associated factor (BAF53a), is involved in diverse cellular processes, such as vesicular transport, spindle orientation, nuclear migration, the cell cycle, and chromatin remodeling[6–9]. Moreover, ACTL6A has been reported to be related to the tumorigenesis of several cancers[10]. For example, Saladi et al. showed that ACTL6A was frequently amplified and highly expressed with TP63 and that they coordinated a regulatory on WWC1 to mediate oncogenic YAP activity, contributing to outcomes of patients with head and neck squamous cell carcinoma[11]. Zeng et al. reported that ACTL6A exhibited protumor function and was an activator of the epithelial-to-mesenchymal transition (EMT) in colon cancer[12]. In addition, Shuai Xiao et al. showed that ACTL6A promoted metastasis and the EMT by activating SOX2/Notch1 signaling in hepatocellular carcinoma[13]. Another study suggested that ACTL6A suppressed p21(Cip1) promoter activity to reduce p21Cip1 protein expression,

[1]Department of General Surgery, Guangdong Provincial Key Laboratory of Colorectal and Pelvic Floor Diseases, The Sixth Affiliated Hospital, Sun Yat-sen University, Guangzhou 510655, China. [2]Biomedical Innovation Center, The Sixth Affiliated Hospital, Sun Yat-sen University, Guangzhou 510655, China. [3]Guangdong Research Institute of Gastroenterology, The Sixth Affiliated Hospital, Sun Yat-sen University, Guangzhou 510655, China. [4]These authors contributed equally: Ziqing Yang, Shaomin Zou. ✉e-mail: kangl@mail.sysu.edu.cn; limh33@mail.sysu.edu.cn; fanglk3@mail.sysu.edu.cn

which was a mechanism to maintain the aggressive phenotype of epidermal squamous cell carcinoma[14]. Furthermore, ACTL6A has been identified as a transcriptional regulator and driving pathways that are of specific benefit to the malignant elements within the tumor[15]. Although ACTL6A has been characterized as an oncogene in many human cancers, its role in GC is unknown, and knowledge of the underlying mechanisms is limited.

Metabolic reprogramming driven by oncogenes allows cancer cells to maintain deregulated cell proliferation, withstand metabolic challenges that are associated with diminished oxygen and nutrients, maintain a dedifferentiated state through alterations in global patterns of gene expression, and corrupt the surrounding microenvironment to actively facilitate tumor growth and dissemination[16–18]. Glutathione (GSH), an abundant antioxidant that

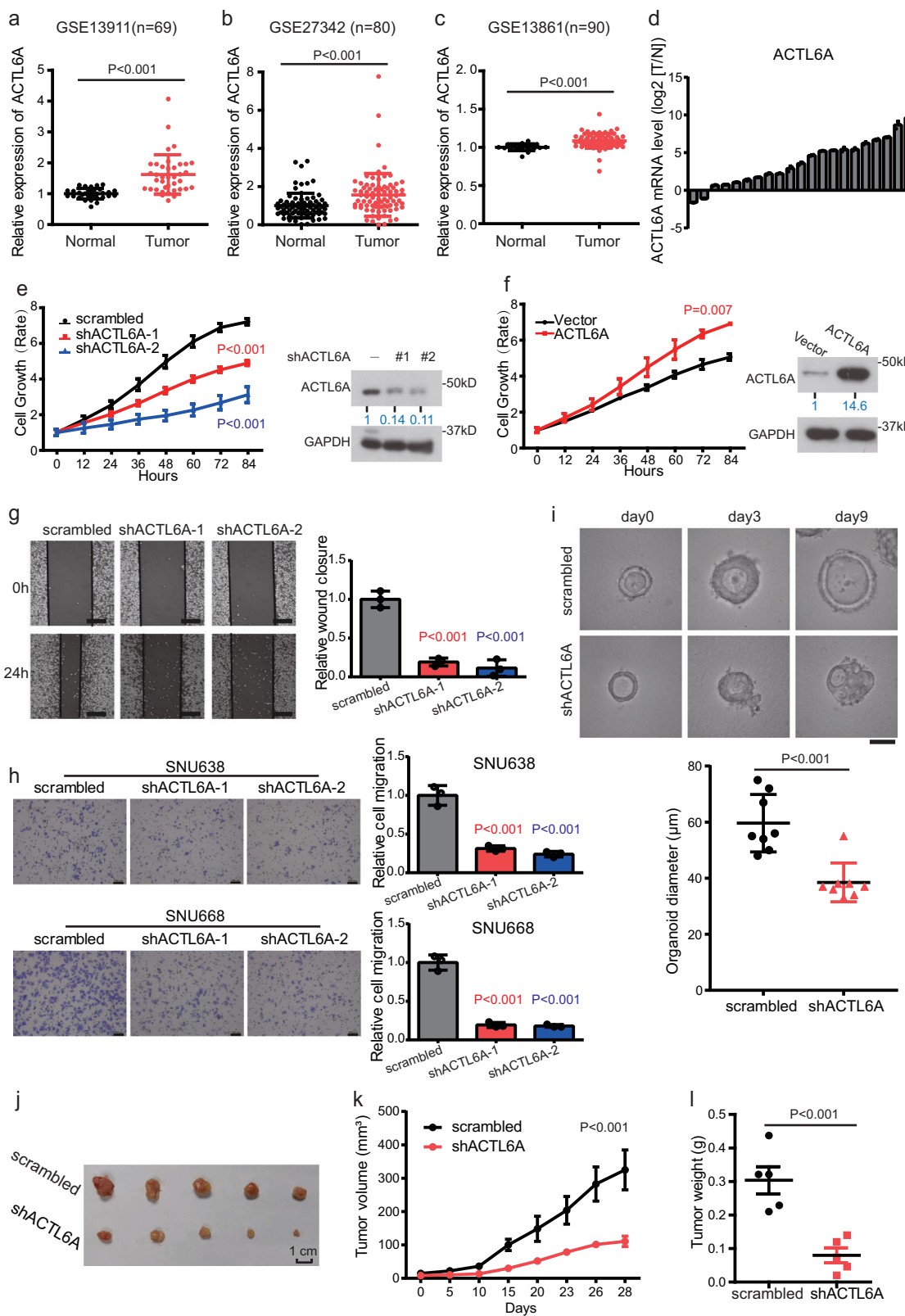

**Fig. 1 | ACTL6A is overexpressed in GC and promotes GC progress. a–c** ACTL6A expression levels in GC and normal tissue from three GEO datasets, GSE13911(**a**), GSE27342 (**b**), and GSE13861 (**c**). The data are presented as the means ± SD (standard deviation). **d** Waterfall plot of the relative ACTL6A mRNA levels from 21 paired samples of GC and normal tissue measured using qRT-PCR. The log2 T/N ACTL6A mRNA levels are presented. Each bar chart represents one case, presented as the means ± SD. **e** Relative cell growth rate of SNU638 cells treated with ACTL6A shRNA and scrambled shRNA. The data are presented as the means ± SEM, $n = 3$ biologically independent experiments. Western blot analysis of ACTL6A protein level is shown. **f** Relative cell growth rate of SNU638 cells treated with flag-ACTL6A and vector. The data are presented as the means ± SEM, $n = 3$ biologically independent experiments. Western blot analysis of ACTL6A protein level is shown. **g** Wound-healing assay of SNU638 cells treated with ACTL6A shRNA or scrambled shRNA.

Scale bar, 200 μm. The data are presented as the means ± SD, $n = 3$ biologically independent experiments. **h** Transwell migration assay of SNU638 and SNU668 cells treated with ACTL6A shRNA or scrambled shRNA. The scale, 100 μm. The data are presented as the means ± SD, $n = 3$ biologically independent experiments. **i** Patient-derived organoids (PDOs) were treated with ACTL6A shRNA or scrambled for 9 days. Representative pictures at day 0, day 3, and day 9. Scale bar, 25 μm. The size of organoids on the 9th day were calculated, presented as the means ± SD, $n = 8$ for each group. **j–l** Xenograft experiment. SNU638 cells ($1 \times 10^6$) treated with ACTL6A shRNA or scrambled were subcutaneously injected into nude mice ($n = 5$ for each group). Images (**j**), tumor volumes (**k**) and tumor weight (**l**) are shown. The data are presented as the means ± SD. *P* values were determined by unpaired two-tailed T test for panels **a–c**, **g–i**, **l** and two-way ANOVA followed by Tukey test for panels **e**, **f**, **k**.

maintains a correct redox balance in cells, is essential for the protection of tumor cells against oxidative stress, such as stress induced by reactive oxygen species (ROS)[19–23]. GSH is synthesized from cysteine, glutamate, and glycine by the ATP-dependent enzyme γ-glutamyl-cysteine (γ-GC) ligase (GCL) synthetase, which is composed of the γ-GC ligase catalytic subunit (GCLC), the γ-GC modifier subunit (GCLM), and GSH synthetase (GSS)[24]. Ogiwara et al. showed that ARID1A-deficient cancer cells expressed low levels of the key cystine transporter SLC7A11 and thus exhibited low basal levels of GSH, which made these cancer cells specifically vulnerable to inhibition of the GSH metabolic pathway[25]. Harris et al. showed that GSH is required for cancer initiation partially because of the upregulated activation of the thioredoxin (TXN) antioxidant pathway and inhibition of both GSH and TXN pathway activation, synergistically inhibiting tumor growth[26]. These studies revealed that GSH metabolism plays an important role in tumor initiation and progression, but the role played by GSH and the mechanism of GSH action in GC remain unclear.

Ferroptosis, a form of regulated cell death that is characterized by the iron-dependent accumulation of lipid hydroperoxides, is morphologically and mechanistically different than apoptosis[27–31]. Ferroptotic cell death is associated with various pathological conditions, including acute kidney injury, hepatocellular degeneration and hemochromatosis, traumatic brain injury, neurodegeneration and carcinogenesis[32–37]. Due to the role played by GSH in regulating redox balance, some molecules involved in GSH metabolism, such as glutathione peroxidase 4 (GPX4), GCLC and the cystine transporter solute carrier family 7 member 11 (SLC7A11), have been reported to inhibit the ferroptosis of cancer cells[38–44]. This evidence indicates that research on ferroptosis-inducing methods is of great significance for cancer treatment. However, the mechanism of cell type-specific ferroptosis sensitivity remains unclear, and the inherent susceptibility to ferroptosis of untreated cancers varies significantly by organ system[39].

In the present study, we systematically showed that ACTL6A functions as a regulator of GSH synthesis and then inhibits ferroptosis of GC cells in vitro and in vivo. Through stable isotope tracing and chromatin-immunoprecipitation (ChIP) assay, we also found that ACTL6A promotes GSH synthesis by upregulating the rate-limiting enzyme GCLC as a co-transcription factor. Furthermore, we revealed that the hydrophobic domain of ACTL6A is essential to the ACTL6A-GCLC-GSH synthesis-ferroptosis axis. Our findings indicated that ACTL6A is a promising target of GC treatment and give an implication for understanding the mechanisms of metabolic reprogramming and the ferroptosis pathway in GC.

## Results

### ACTL6A is overexpressed in GC and promotes GC progress
First, analysis of mRNA expression data obtained from the GC datasets GSE13911, GSE27342, GSE13861 in the Gene Expression Omnibus (GEO) demonstrated that the ACTL6A expression level was higher in cancer tissues than in normal tissues (Fig. 1a–c and Supplementary Fig. 1a).

Then we validated that ACTL6A mRNA expression levels were upregulated in GC through analysis of 21 paired samples of GC tissue and adjacent normal mucosa revealed (Fig. 1d).

Then, we further explored the role played ACTL6A in GC tumor progress. ACTL6A knockdown (KD) via short hairpin RNA (shRNA) transfection exerted a marked negative effect on GC cell proliferation, while overexpression of ACTL6A exerted a positive effect on GC cell proliferation (Fig. 1e, f and Supplementary Fig. 1b). Moreover, ACTL6A overexpression relieved the inhibited GC cell proliferation caused by ACTL6A expression knockdown (Supplementary Fig 1c, d). Introduction of shRNA to knock down ACTL6A expression inhibited the wound healing and metastatic ability of GC cells (Fig. 1g, h). Furthermore, ACTL6A KD via shRNA inhibited patient-derived organoid (PDO) growth, and organoids with ACTL6A KD had a compact morphology with no lumen (Fig. 1i). As shown in Fig. 1j–l, knocking down ACTL6A significantly reduced SNU638 xenograft growth.

In conclusion, ACTL6A is highly expressed in GC, and its positive impact on GC cell growth contributes to its oncogenic effect.

### ACTL6A reprograms GSH metabolism to maintain GC malignant progression
To identify the roles played by ACTL6A in GC, we performed an RNA array in ACTL6A-KD SNU638 cells. We then performed a gene set enrichment analysis (GSEA) to determine the association between the expression of ACTL6A and signaling pathways in the RNA array results, which indicated significant differences between ACTL6A-KD cells and control cells in multiple metabolism-related pathways, among which GSH metabolism ranked as one of the most highly correlated pathways (Fig. 2a and Supplementary Fig. 2a). This result could be repeated in two other datasets of GC, GSE15459 and GSE27342 (Fig. 2b, c).

GSH, an antioxidant, protects cancer cells from oxidative stress via the GSH/GSSG cycle, which decreases intracellular ROS levels and transforms NADPH to NADP+ (Fig. 2d), thereby promoting tumor growth[16]. We verified the impact of ACTL6A on GSH synthesis and confirmed that ACTL6A KD led to reduced expression of certain genes (GCLC, GPX2, GPX4, SLC7A11, SLC1A5, GSS, GLS, etc.) involved in the GSH synthesis pathway (Supplementary Fig. 2b). We also showed that GSH/GSSG and NADP + /NADPH ratios were reduced in ACTL6A-KD cells (Fig. 2e, f and Supplementary Fig. 2c). Additionally, as detected with a Seahorse XF analyzer, we found that the oxygen consumption rate (OCR) and extracellular acidification rate (ECAR) were both decreased in ACTL6A-KD cells (Supplementary Fig 2d, e). Next, we detected the levels of DCFH-DA and found that, compared to control cells, the ROS levels were significantly higher in ACTL6A-KD cells with or without $H_2O_2$ treatment (Fig. 2g and Supplementary Fig 2f).

Furthermore, we found that N-acetylcysteine (NAC), an antioxidant, reestablished ACTL6A-KD cell proliferation (Fig. 2h, i and Supplementary Fig. 2g). Treatment with NAC restored the sizes of ACTL6A-KD PDOs that had been decreased, and enabled organoids regained a cystic structure (Fig. 2j). To determine the in vivo functional

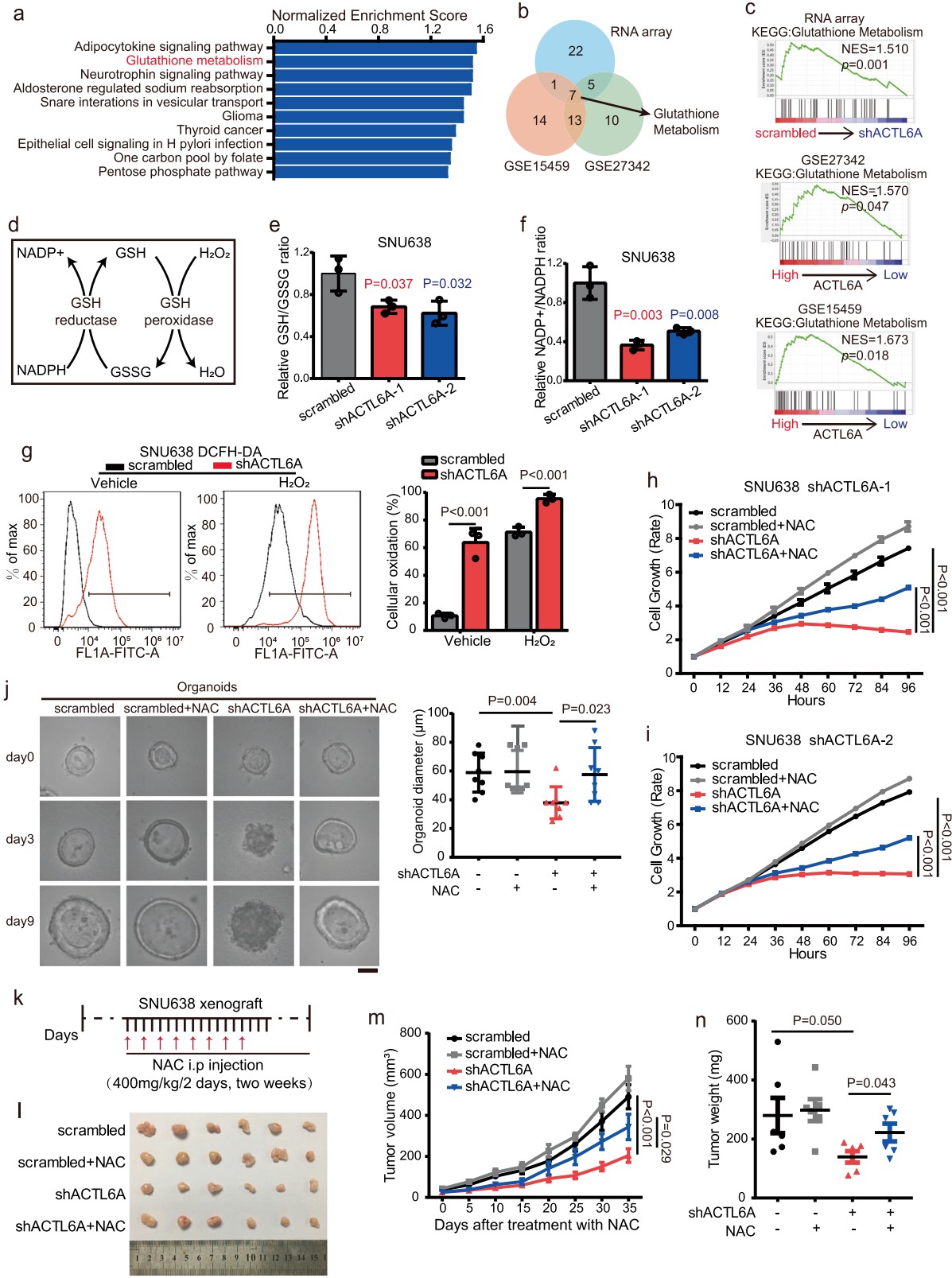

contribution of ACTL6A to tumor progress, we performed GC xenograft mouse model experiments. Knocking down ACTL6A expression with shACTL6A significantly reduced tumor growth, and NAC treatment (intraperitoneal (i.p.) injection, 200 mg/kg/day) significantly reversed this effect (Fig. 2k–n).

In conclusion, ACTL6A accelerates GSH synthesis and then reduces ROS levels to support GC tumor growth.

## ACTL6A inhibits ferroptosis of GC cells

Because ferroptosis is linked to ROS and GSH regulation, we sought to determine whether ACTL6A inhibits the ferroptosis of GC cells. We found that cell viability was significantly restored by ferroptosis inhibitor ferrostatin-1 (Fer-1) treatment in ACTL6A-ablated cells but not by apoptosis inhibitor ZVAD-FMK or necroptosis inhibitor necrostatin-1 treatment (Fig. 3a and Supplementary Fig 3a). Furthermore, we

**Fig. 2 | ACTL6A reprograms GSH metabolism to maintain GC malignant progression. a** Gene categories significantly ($P \leq 0.05$) enriched for genes deregulated in KEGG pathways, owing to knockdown of ACTL6A in SNU638 cells. **b** Venn diagram illustrating the overlap among the top 35 enriched pathways of RNA array and two GC databases. **c** An enrichment analysis of gene sets available from RNA array and two GC databases revealed that ACTL6A expression is positively correlated with Glutathione Metabolism. **d** Schematic of GSH/GSSG cycle that affects oxidative stress and NADPH/NADP+ ratio. **e** Measurement of relative GSH/GSSG ratio in SNU638 cells treated with ACTL6A shRNA and scrambled shRNA. The data are presented as the means ± SD, $n = 3$ biologically independent experiments. **f** Measurement of relative NADP + /NADPH ratio in SNU638 cells treated with ACTL6A shRNA and scrambled shRNA. The data are presented as the means ± SD, $n = 3$ biologically independent experiments. **g** Relative DCFH-DA fluorescence intensity measured by flow cytometry, SNU638 cells are treated with ACTL6A shRNA and scrambled shRNA and cultured with or without 50 µM $H_2O_2$ for 24 h.

Data are presented as the means ± SD, $n = 3$ biologically independent experiments. **h, i** Relative cell growth rate of SNU638 cells treated with ACTL6A shRNA and scrambled shRNA and cultured with or without 100 µM NAC. The data are presented as the means ± SEM, $n = 3$ biologically independent experiments. **j** PDOs were treated with ACTL6A shRNA or scrambled shRNA and cultured with or without 100 µM NAC for 9 days. Representative pictures at day 0, day 3, and day 9 after being treated with NAC were shown. Scale bars represent 25 µm. The size of organoids on the 9th day was calculated, presented as the means ± SD, $n = 8$ for each group. **k–n** SNU638 cells ($1 \times 10^6$) treated with ACTL6A shRNA or scrambled shRNA were transplanted to nude mice ($n = 6$ for each group). Mice were treated with or without NAC (**k**). Images (**l**), tumor volumes (**m**) and tumor weight (**n**) are shown. The data are presented as the means ± SD. $P$ values were determined by unpaired two-tailed T test for panels **e–g, j, n**, and two-way ANOVA followed by Tukey test for panels **h, i, m**.

observed that ACTL6A suppression sensitized GC cells to $H_2O_2$, erastin, and buthionine sulfoximine (BSO), which are ferroptosis inducers, but not to compounds (e.g., etoposide and doxorubicin) that inhibit cell growth through other mechanisms (Fig. 3b–f). A hallmark of ferroptosis is the accumulation of lipid peroxide; therefore, we performed C11-BODIPY staining in cells expressing shACTL6A and found that the lipid peroxidation levels had increased and that the phenotypes were more obvious in cells treated with erastin and could be changed by Fer-1 treatment (Fig. 3g and Supplementary Fig 3c). Furthermore, treatment with Fer-1 relieved the inhibition of PDO growth that had been induced by ACTL6A KD, and enabled organoids regained a cystic structure (Fig. 3h). Importantly, by immunohistochemical (IHC) staining of xenograft tumor tissues, we found that knocking down ACTL6A expression significantly decreased the level of Ki67, a proliferation marker, and increased that of 4-hydroxy-2-noneal (4-HNE)[45], a lipid peroxidation marker (Fig. 3i). In addition, knocking down ACTL6A increased the expression levels of prostaglandin-endoperoxide synthase 2 (PTGS2) (Supplementary Fig 3b), which is a ferroptosis marker[30,46].

Therefore, inhibition of GSH biosynthesis via ACTL6A ablation promotes the induction of ferroptosis and sensitizes cells to oxidative stress.

## ACTL6A impacts GSH de novo synthesis mainly by upregulating γ-glutamyl-cysteine synthesis

To further assess the mechanism by which ACTL6A regulates GSH metabolism, we investigated the metabolic fate of U-$^{13}$C glucose by liquid chromatography with mass spectrometry (LC−MS), which produces γ-GC containing two $^{13}$C atoms via GSH de novo synthesis pathway, GSH containing two $^{13}$C atoms via GSH de novo synthesis or the SGOC pathway, or four $^{13}$C atoms via both pathways (Fig. 4a). As a result, no significant change in the total contributions of U$^{13}$C-glucose to glutamate, serine or glycine was observed (Fig. 4b–d). Nevertheless, analysis of the mass isotopomer contribution of U-$^{13}$C glucose to γ-GC and GSH showed that the levels of M + 2-labeled [$^{13}$C] -γ-GC and M + 2- and M + 4-labeled [$^{13}$C]-GSH were reduced in shACTL6A-expressing cells (Fig. 4e, f). In addition, we investigated the metabolic fate of U-$^{13}$C glutamine, which produces γ-GC and GSH containing five $^{13}$C atoms via GSH de novo synthesis (Fig. 4g). Consistently, there was no significant change in the total contributions of U$^{13}$C-glucose to glutamate (Fig. 4h). Moreover, analysis of mass isotopomer contribution of U-$^{13}$C glutamine to γ-GC and GSH confirmed the reduction in M + 5-labeled [$^{13}$C]-γ-GC and M + 5-labeled [$^{13}$C]-GSH in the shACTL6A-expressing cells. (Fig. 4i, j). As expected, the total intracellular γ-GC and GSH pools were significantly decreased in cells expressing shACTL6A (Supplementary Fig 4a, b). Additionally, we detected metabolites of SNU638 xenograft tumor tissues by LC−MS and found that γ-GC and GSH expression was downregulated after ACTL6A expression was knocked down (Supplementary Fig. 4c, d).

Together, these findings enabled us to conclude that ACTL6A impacted GSH de novo synthesis mainly by regulating the synthesis of γ-GC.

## ACTL6A regulates GCLC in transcriptional level via NRF2

The synthesis of γ-GC is catalyzed by GCL, which is the rate-limiting enzyme in the GSH biosynthesis pathway. GCL is a heterodimer comprising a catalytic subunit (GCLC) and a regulatory subunit (GCLM)[47]. We showed that the GCLC mRNA expression level was downregulated by ACTL6A KD and was upregulated when ACTL6A was overexpressed, but the GCLM expression was not changed (Supplementary Fig 5a, b). Knocking down ACTL6A also decreased the GCLC protein expression level in GC cells (Supplementary Fig 5c). In addition, knocking down ACTL6A inhibited GCLC expression in xenograft tissues (Fig. 5a and Supplementary Fig. 5d, e).

We then overexpressed GCLC in cells with ACTL6A knocked down and found that GCLC reversed the inhibition of cell proliferation (Fig. 5b and Supplementary Fig. 5f). As detected by DCFH-DA staining, GCLC overexpression reestablished cellular ROS levels in cells expressing shACTL6A, phenotypes remained significant when treated with $H_2O_2$ and was reversed by NAC (Fig. 5c and Supplementary Fig. 5g). Additionally, C11-BODIPY staining indicated that GCLC overexpression restored lipid peroxidation levels in cells expressing shACTL6A, phenotypes remained significant when treated with erastin and was reversed by Fer-1 (Fig. 5d and Supplementary Fig. 5h).

ACTL6A, a member of the BAF chromatin remodeling complex, has been reported to be a transcription factor[48]. Therefore, to explore how ACTL6A regulates GCLC, we assessed genome-wide ACTL6A binding by performing a chromatin immunoprecipitation high-throughput sequencing (ChIP-seq) analysis of endogenous ACTL6A in GC cells. The analysis of enriched loci (peaks) indicating ACTL6A binding revealed, as expected, a prominent peak (GRCh38:6:53545016:53545243) near the transcriptional start site (TSS) of GCLC (Fig. 6a), the entire genomic loci of GCLC was showed in Supplementary Fig. 6a. Besides, we conducted an experiment of anti-BRG1 (the catalytic subunit of the SWI/SNF chromatin-remodeling complex) ChIP-seq. The results showed that BRG1 could bind to the TSS of GCLC, and the peak was overlapped with that of anti-ACTL6A ChIP-seq (Supplementary Fig. 6a). Through the website JASPAR (http://jaspar.genereg.net/), we observed that, among the peaks near the TSS in GCLC, a sequence was highly homologous to the NRF2-binding motif (Fig. 6b). Nuclear factor (erythroid-derived 2)-like 2 (NFE2L2, NRF2) is a transcription factor that governs the antioxidant pathway, targeting GSH metabolism-related genes, including GCLC, GCLM, and SLC7A11[41,49]. Therefore, we assumed that ACTL6A might work cooperatively with NRF2 to regulate GCLC expression. To test our hypothesis, we first detected GCLC expression and found that when NRF2 was knocked down, the

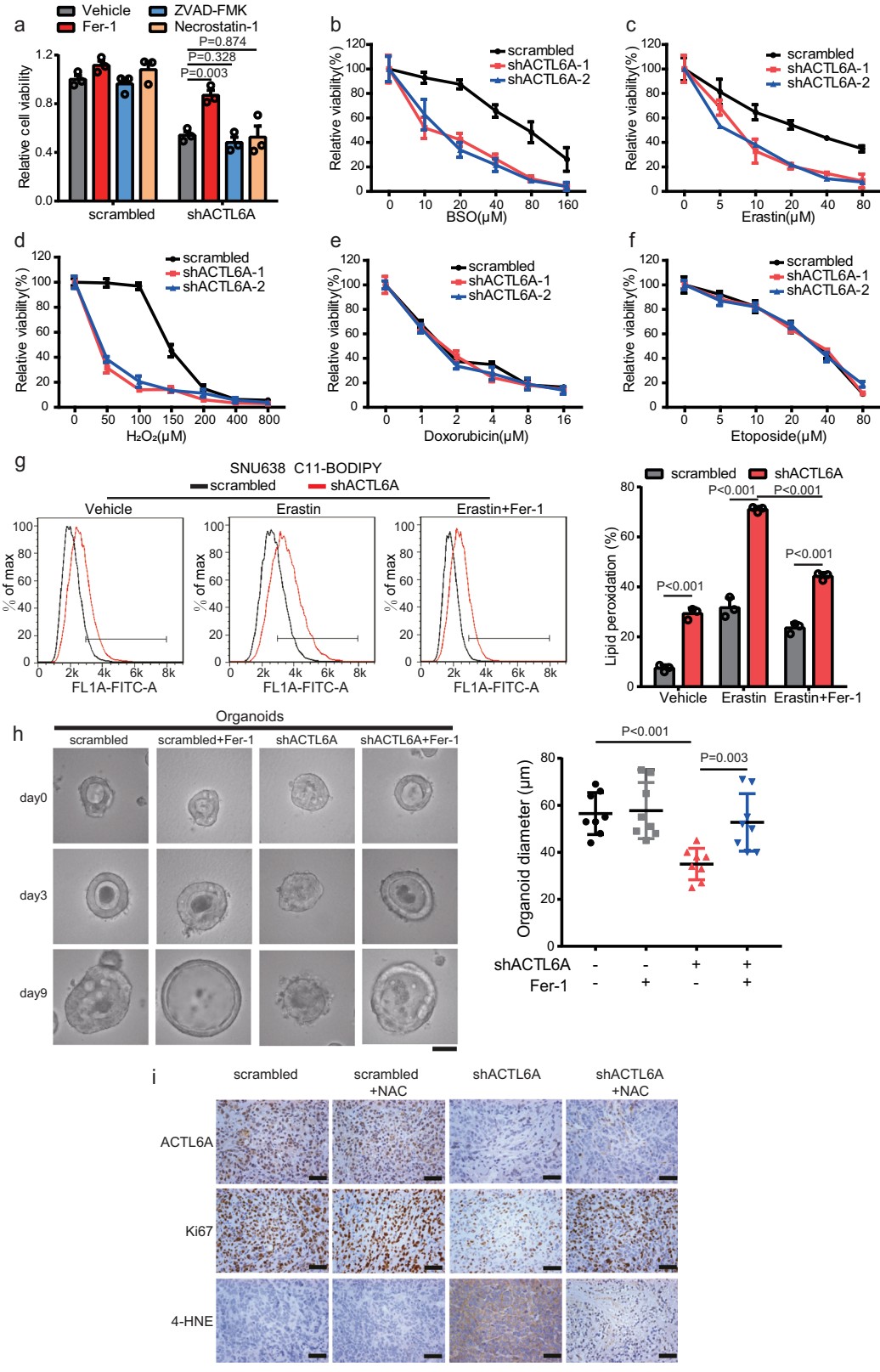

inhibition of GCLC expression by ACTL6A KD was diminished; similarly, when ACTL6A was knocked down, NRF2 KD could not decrease GCLC expression (Fig. 6c). The anti-NRF2 ChIP-seq results showed that NRF2 could also bind to the promoter of GCLC (Supplementary Fig. 6a). Next, we performed ChIP followed by qRT-PCR and found that both ACTL6A and NRF2 interacted with the predicted binding site near the TSS of GCLC (Fig. 6d, e and Supplementary Fig. 6b, c).

Furthermore, the interaction of ACTL6A with GCLC promoter was weakened by NRF2 KD, and likewise, the interaction of NRF2 with GCLC promoter was weakened by ACTL6A KD (Fig. 6f, g and Supplementary Fig. 6d, e). The interaction of NRF2 and ACTL6A in GC cells was confirmed by co-immunoprecipitation (co-IP) (Fig. 6h, i and Supplementary Fig. 6f, g). Knocking down ACTL6A had no impact on NRF2 expression (Supplementary Fig 6h).

**Fig. 3 | ACTL6A inhibits ferroptotic cell death. a** Viability of SNU638 cells treated with ACTL6A shRNA or scrambled shRNA after 24 h cultured with or without 1 μM Fer-1, 2 μM necrostatin-1, 5 μM ZVAD-FMK. The data are presented as the means ± SD, $n = 3$ biologically independent experiments. **b**–**f** Viability of SNU638 cells treated with ACTL6A shRNA or scrambled shRNA after 24 h cultured with or without varying concentrations of BSO (**b**), erastin (**c**), H$_2$O$_2$ (**d**), doxorubicin (**e**), and etoposide (**f**). The data are presented as the means ± SD, $n = 3$ biologically independent experiments. **g** Relative C11-BODIPY fluorescence measured by flow cytometry of SNU638 cells treated with ACTL6A shRNA or scrambled shRNA and cultured with either erastin (10 μM), Fer-1 (1 μM) or both for 24 h. The percentages of lipid peroxidation are presented as the means ± SD, $n = 3$ biologically independent experiments. **h** PDOs were treated with ACTL6A shRNA or scrambled shRNA and cultured with or without 5 μM Fer-1 for 9 days. Representative pictures at day 0, day 3, and day 9 after being treated with Fer-1 were shown. Scale bars represent 25μm. The size of organoids on the 9th day were calculated. Data are presented as the means ± SD, $n = 8$ for each group. **i** Immunohistochemical (IHC) staining for ACTL6A, Ki-67 and 4-HNE in xenograft tissues derived from ACTL6A-knockdown SNU638 cells in mice treated with or without NAC. Scale bars represent 50 μm. *P* values were determined by unpaired two-tailed T test for panels **a**, **g**, **h**.

Altogether, NRF2 is a transcriptional factor of GCLC, and ACTL6A co-works with NRF2 and affects NRF2 binding to the promoter of GCLC, thereby regulates GCLC expression at the transcriptional level.

### The HR domain of ACTL6A is essential in regulating GCLC and ferroptosis

ACTL6A interactions with cofactors are essential for oncogenesis. It has been reported that some domains of ACTL6A suggest that it is a member of the actin-related gene family, which is important in protein–protein interactions[15]. In the ΔDNBL (DNase 1-binding loop) mutant (Δ39–67), the ACTL6-specific region at the position equivalent to that of the DNase 1-binding domain of β-actin was deleted. In the ΔCC (cation coordination) mutant (Δ171–179), residues equivalent to those coordinating Ca$^{2+}$ or Mg$^{2+}$ ions in β-actin were deleted. In the ΔNBC (nucleotide-binding cleft) mutant (Δ233–255), a region equivalent to that in the nucleotide-binding cleft of β-actin was deleted. In the ΔHR (hydrophobic region) mutant (Δ319–324), a hydrophobic region β-actin region that may mediate protein–protein interactions, was deleted[15]. These four deletion mutants were constructed and transfected into ACTL6A-KD cells (Fig. 7a). We found that wild-type (WT) ACTL6A and the ΔDNBL, ΔCC, and ΔNBC ACTL6A mutants reversed the inhibition of cell proliferation and the reduced GCLC protein expression induced by knocking down ACTL6A expression, but the ACTL6A HR-deleted mutant did not show this restorative effect (Fig. 7b, c and Supplementary Fig. 7a, b). Moreover, increasing the dose of the ACTL6A HR-deleted mutant failed to reverse the reduction in GCLC mRNA expression caused by knocking down ACTL6A expression (Fig. 7d). This finding indicates that the HR domain of ACTL6A plays an essential role in GCLC regulation; therefore, we performed co-IP and found that ACTL6A ΔHR mutant did not bind to NRF2 (Fig. 7e and Supplementary Fig. 7c). ChIP assays showed that, in contrast to WT ACTL6A, the ACTL6A HR-deletion mutant exhibited no restorative effect on the binding of NRF2 with the promoter of GCLC (Fig. 7f and Supplementary Fig. 7d).

We reasoned that the HR domain of ACTL6A exerts an impact on ferroptosis. Therefore, we preformed DCFH-DA and C11-BODIPY staining followed by flow cytometry and found that WT ACTL6A reversed the increase in ROS levels (in cells treated with or without H$_2$O$_2$ or both H$_2$O$_2$ and NAC) and lipid peroxidation levels (in cells treated with or without erastin or both erastin and Fer-1), while ACTL6A HR-deleted mutant did not reverse these high levels (Fig. 7g, h and Supplementary Fig. 7e, f).

### Clinical relevance of the ACTL6A-GCLC-GSH metabolism axis in ferroptosis of GC cells

Next, we sought to determine the clinical relevance of ACTL6A in regulating GCLC and inhibiting ferroptosis in a patient-derived xenograft (PDX) model. To this end, after ACTL6A expression levels were detected (Fig. 8a), we implanted fresh primary tumor samples resected from GC patients into immunocompromised mice and then injected PBS or the GCLC inhibitor BSO (i.p. injection, 750 mg/kg/day)[25,50] to the mice (Fig. 8b). As expected, administration of BSO in the established high ACTL6A-expressing PDX tumors (cases 1 and 3) attenuated tumor progression. In contrast, GCLC inhibition exerted a minimal impact on

the growth of PDX tumors with low ACTL6A expression (cases 2 and 4) tumors (Fig. 8c–f and Supplementary Fig. 8a). And there comes with high GCLC expression in ACTL6A-high PDX tumors (Fig. 8a). In addition, BSO decreased the intensity of Ki67 staining and increased that of 4-HNE staining in tumors with high ACTL6A expression, but this trend was not observed in PDX tumors with low ACTL6A expression (Supplementary Fig 8b). As detected by DCFH-DA and C11-BODIPY staining of frozen sections of the PDX tumors, the ROS levels and lipid peroxidation levels of the PDX tumors with high ACTL6A expression were lower, and they were significantly increased by BSO treatment; in contrast, those of PDX tumors with low ACTL6A expression were higher and were not further increased by BSO treatment (Fig. 8g–j). These results indicated that tumors with high ACTL6A expression, which rely more heavily on GSH metabolism than those with low ACTL6A expression, are more sensitive to ferroptosis inducers or GCLC inhibitors.

Furthermore, we investigated the metabolic fate of U-$^{15}$N glutamine in vivo by LC-MS, which produces glutathione containing one $^{15}$N atom via GSH de novo synthesis (Fig. 8k). We found that administration of BSO to established PDX tumors with high ACTL6A expression decreased the total GSH level but exerted no impact on the glutamine or glutamate level (Fig. 8l). Along with the process of γ-GC synthesis was blocked by BSO, the total contributions of U-$^{15}$N glutamine to GSH was reduced in the established high ACTL6A-expressing PDX tumors, and the levels of [$^{15}$N]-labeled glutamine and glutamate were thus increased (Fig. 8m). In addition, the ratio of GSH/Gln was reduced by BSO treatment in the established PDX tumors with high ACTL6A expression (Fig. 8n). However, administration of BSO seemed to have little influence on the process of GSH synthesis in the established PDX tumors with low ACTL6A expression (Supplementary Fig. 8c–e). These results supported the hypothesis that ACTL6A impacts GSH de novo synthesis mainly by regulating GCLC expression.

Analysis of paired samples of GC tissue and adjacent normal mucosa revealed that the GC samples exhibited high ACTL6A and GCLC protein expression levels (Supplementary Fig. 8f). We further evaluated the clinical relevance of these findings by analyzing the expression of ACTL6A and GCLC with a tissue microarray containing 184 GC tissue specimens. A Kaplan–Meier analysis showed that high ACTL6A and GCLC levels were correlated with poor overall survival (Fig. 8o). Notably, GCLC expression was significantly positively correlated with ACTL6A in GC, as indicated by the analysis of the 184 GC samples in the microarray (Fig. 8p and Table 1). The clinicopathological features of GC patients are shown in Supplementary Data 1.

## Discussion

Collectively, these results suggested that ACTL6A, which contains a hydrophobic region, cooperates with NRF2 to regulate GCLC expression at the transcriptional level and then promotes GSH synthesis and inhibits the ferroptosis of GC cells. (Fig. 9).

The SWI/SNF (BAF) complex, an ATP-dependent chromatin remodeling complex in the nucleus that endows DNA with the capacity for replication and enables selective gene expression and DNA repair and recombination, is frequently mutated in human cancers and thus exerts a great impact on tumorigenesis[51]. ACTL6A, a member of

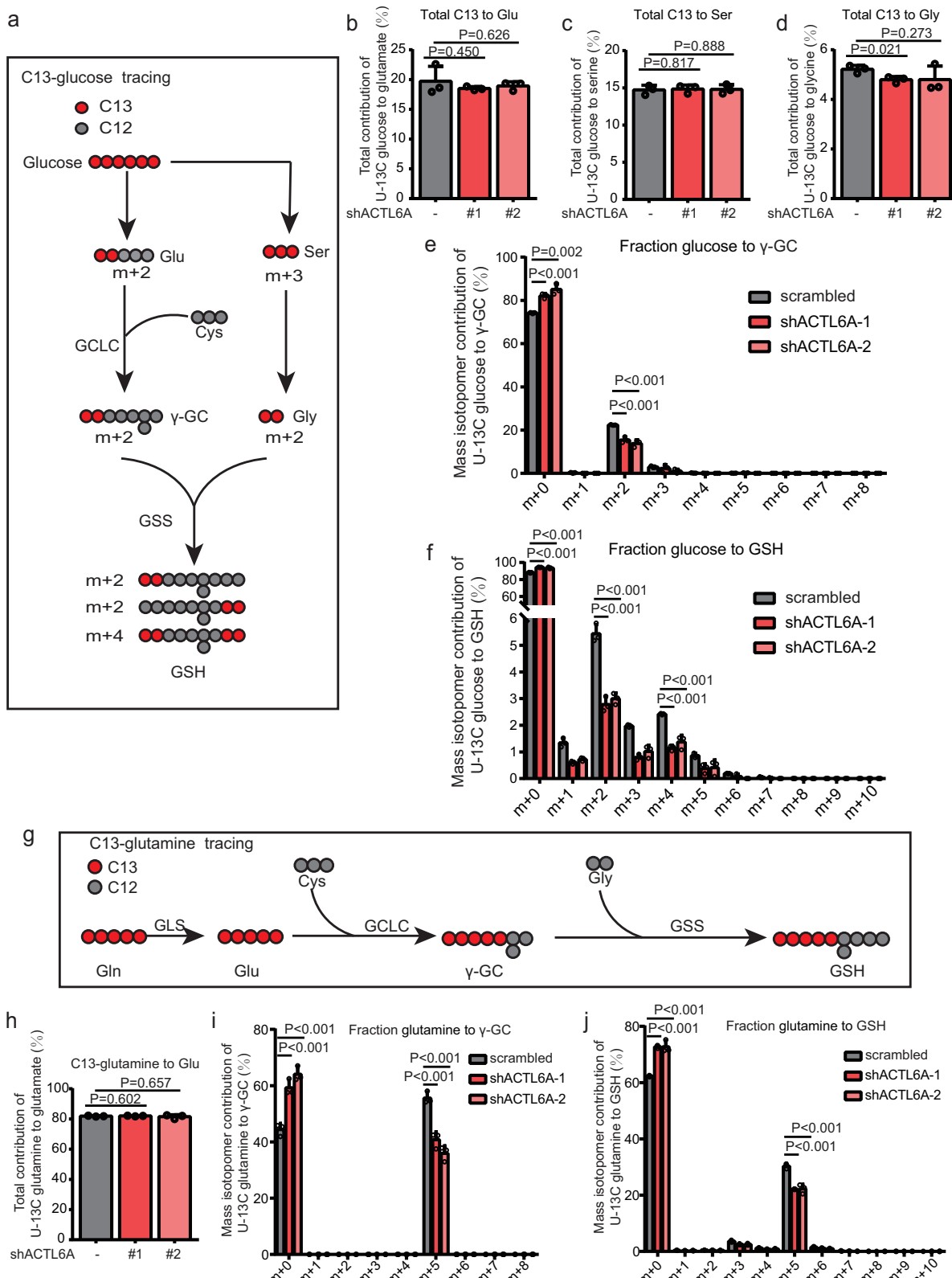

SWI/SNF complex, has been reported to be involved in various cancers, including squamous cell carcinoma[11,14,52], colon cancer[12], osteosarcoma[53], glioma[10,54], and hepatocellular carcinoma[13]. However, its functions and mechanisms in cancers, especially in GC, are poorly understood. This study uncoverd the oncogenic role of ACTL6A in GC and its mechanisms, which might be potentially translatable to diagnostic methods or therapies for GC patients.

Metabolic reprogramming is regarded as a hallmark of cancers because of the high energy/anabolism needs for cancer cell growth[16]. The role of GSH metabolism in cell differentiation, proliferation and apoptosis has been previously shown to lead to cancer progression and chemoresistance[21]. Nevertheless, the ability to selectively target GSH metabolic components in therapeutic applications remains a challenge; therefore, a more in-depth exploration into the mechanisms

**Fig. 4 | ACTL6A impacts GSH de novo synthesis mainly by upregulating γ-glutamyl-cysteine synthesis. a** Schematic metabolic map of [U-¹³C] glucose-labeled GSH de novo synthesis. Glu, glutamate; Cys, cysteine; Ser, serine; Gly, glycine. γ-GC, γ-glutamyl-cysteine; GSH, glutathione. **b–d** Incorporation of carbon atoms from [U-¹³C] glucose into glutamate (**b**), serine (**c**), and glycine (**d**) in scrambled and ACTL6A-KD SNU638 cells. Data are presented as the means ± SD, $n = 3$ biologically independent experiments. **e** Fractional contribution of carbon atoms from [U-¹³C] glucose into the γ-GC (m + 0 to m + 10) isotopomers in scrambled and ACTL6A-KD SNU638 cells. Data are presented as the means ± SD, $n = 3$ biologically independent experiments. **f** Fractional contribution of carbon atoms from [U-¹³C] glucose into the GSH (m + 0 to m + 10) isotopomers in scrambled and ACTL6A-KD SNU638 cells. Data are presented as the means ± SD, $n = 3$ biologically

independent experiments. **g** Schematic metabolic map of [U-¹³C glutamine]-labeled GSH de novo synthesis. Gln glutamine, Glu glutamate, Cys cysteine; Gly, glycine, γ-GC γ-glutamyl-cysteine, GSH glutathione. **h** Incorporation of carbon atoms from [U-¹³C] glutamine into glutamate in scrambled and ACTL6A-KD SNU638 cells. Data are presented as the means ± SD, $n = 3$ biologically independent experiments. **i** Fractional contribution of carbon atoms from [U-¹³C] glutamine into the γ-GC (m + 0 to m + 10) isotopomers in scrambled and ACTL6A-KD SNU638 cells. Data are presented as the means ± SD, $n = 3$ biologically independent experiments. **j** Fractional contribution of carbon atoms from [U-¹³C] glutamine into the GSH (m + 0 to m + 10) isotopomers in scrambled and ACTL6A-KD SNU638 cells. Data are presented as the means ± SD, $n = 3$ biologically independent experiments. *P* values were determined by unpaired two-tailed T test for panels **b–f**, **h–j**.

underlying the regulation of the GSH synthesis pathway is needed. In our study, by performing metabolite pool and isotopomer labeling analyses, we found that ACTL6A increased GSH de novo synthesis mainly by regulating the synthesis of γ-GC, the rate-limiting step catalyzed by GCLC. The present finding will help to understand the metabolic reprogramming in cancers, especially in GC.

In a previous study[55], they made definition criteria of high vs. low target molecule (MAP3K8) in PDX. MEK inhibitors markedly reduced tumor growth in high-MAP3K8 PDX models, compared with low-MAP3K8 PDX models. Similarly, PDX experiments in this study showed that tumors with high ACTL6A expression were more sensitive to administration of BSO, a GCLC inhibitor. There might be other pathways/genes account for the differences. For example, pathways involved in glutathione metabolism or GCLC regulation might also be modified in the cases, which would affect the sensitivity of PDX tumors to BSO. Besides, other pathways/genes that influence redox balance or ferroptosis might also account for the differences. Based on the function of ACTL6A on GCLC regulating, we believe that the expression of ACTL6A plays an important role in it. This result provides strong evidence that the sensitivity of GC to inhibition of the GSH synthesis by targeting GCLC is defined by ACTL6A. BSO, or other drugs targeting GCLC, is a candidate drug for GCs with high ACTL6A expression.

ROS increasing is correlated with abnormal cancer cell growth. However, ROS may have a cytotoxic effect to cancer cell if the ROS reaches a certain threshold level[56]. Generally, high ROS occur in GC cells due to H. pylori infections, EBV infections, or other molecular mechanisms[57]. Therefore, ACTL6A might promote GC cells proliferation via reducing ROS levels. Because of the role played by GSH in regulating redox balance, GSH metabolism has been previously reported to inhibit ferroptosis of cancer cells[58,59]. Ferroptosis represents a unique form of regulated cell death, and many of its physiological roles have yet to be defined[28]. Further definition of the genotype-selective activity in ferroptosis in cancer and the mechanisms involved are important to guide ferroptosis-based therapeutic intervention[58,60]. In our study, we systematically demonstrated that ACTL6A inhibited GC cell ferroptosis by promoting GSH synthesis through upregulating GCLC. This finding gives an improved understanding of the role played by ferroptosis in cancer, which might generate opportunities for diagnostics and therapeutic interventions for GCs.

NRF2 is a transcription factor that governs the antioxidant pathway by targeting GSH metabolism-related genes[41,49]. In our study, we confirmed that ACTL6A is a cotranscription factor in the NRF2-dependent regulation of GCLC. Due to the fact that ACTL6A did not regulate NRF2, we could reach the conclusion that ACTL6A functions in conjunction with NRF2 to regulate GCLC expression at the transcriptional level. Since ACTL6A interactions with cofactors are essential for oncogenesis, a thorough understanding of the structural components that enable its cotranscriptional effects is critical. Therefore, we constructed four deletion mutants of ACTL6A, including ΔDNBL, ΔCC, ΔNBC, and ΔHR, that had been previously reported to be

involved in protein–protein interactions[15]. Notably, we found that the HR domain in ACTL6A was essential for regulating GCLC expression. We also confirmed that the HR domain of ACTL6A exerted an impact on redox balance and ferroptosis. Based on the results reported here, we have a thorough understanding of the mechanism about ACTL6A regulating GCLC, GSH metabolism, and ferroptosis. More significantly, we were able to link ACTL6A with NRF2-related antioxidant pathway, and to link NRF2 with ferroptosis.

## Methods

All mouse experiments were approved by the Animal Ethical and Welfare Committee of the Sixth Affiliated Hospital of Sun Yat-sen University carried out following their legal requirements. The collection and use of clinical samples were in accordance with research ethics board approval from the Sixth Affiliated Hospital of Sun Yat-sen University Review Board.

### Patients and tissue samples

We obtained paraffin-embedded samples of primary gastric adenocarcinomas (prepared as Tissue Microarray, TMA) from the Department of Surgery at the Sixth Affiliated Hospital of Sun Yat-sen University. The original immunohistochemistry slides were scanned by Aperio Versa (Leica Biosystems) which captured digital images of the immunostained slides. The Genie calculates an H-score for regions selected by the pathologist. The receiver operating characteristic curve was used to define the cut-off point. All samples were collected with the patients' written informed consent and approval from the Sixth Affiliated Hospital of Sun Yat-sen University Review Board (ethics code: 2021ZSLYEC-100).

### Cell culture and transfection

Gastric cancer cell lines SNU638 (KCLB No.00638), SNU216 (KCLB No.00216), and SNU668 (KCLB No.00668) cells were obtained from KCLB (Korean Cell Line Bank), and HEK293T (CRL-3216) cells were obtained from ATCC (American Type Culture Collection). All the cells were cultured at 37 °C and 5% $CO_2$. SNU638, SNU216, and SNU668 cells were maintained in RPMI 1640 medium (RPMI) supplemented with 10% (v/v) fetal bovine serum (FBS). HEK293T cells were cultured with Dulbecco's modified Eagle's medium (DMEM) with 10% FBS. All transient transfections of plasmids into cell lines followed the standard protocol for Polyethylenimine Linear (PEI) Transfection Reagent (24765-1, Polyysciences Inc).

### shRNA knockdown of ACTL6A and NRF2

We screened four hairpin shRNAs targeting CDS (Coding sequence) of human ACTL6A transcripts and found two independent sequences that reduced mRNA levels by > 70%. These shRNAs were in the pLKO.1 vector (shACTL6A-1 and shACTL6A-2). Besides, a shRNA targeting 3'-UTR of human ACTL6A transcripts was constructed in the pLKO.1 vector (shACTL6A). For NRF2, we constructed four hairpin shRNAs and one of the most efficient one was used (shNRF2). The targeting sequences used were show in Supplementary Table 1.

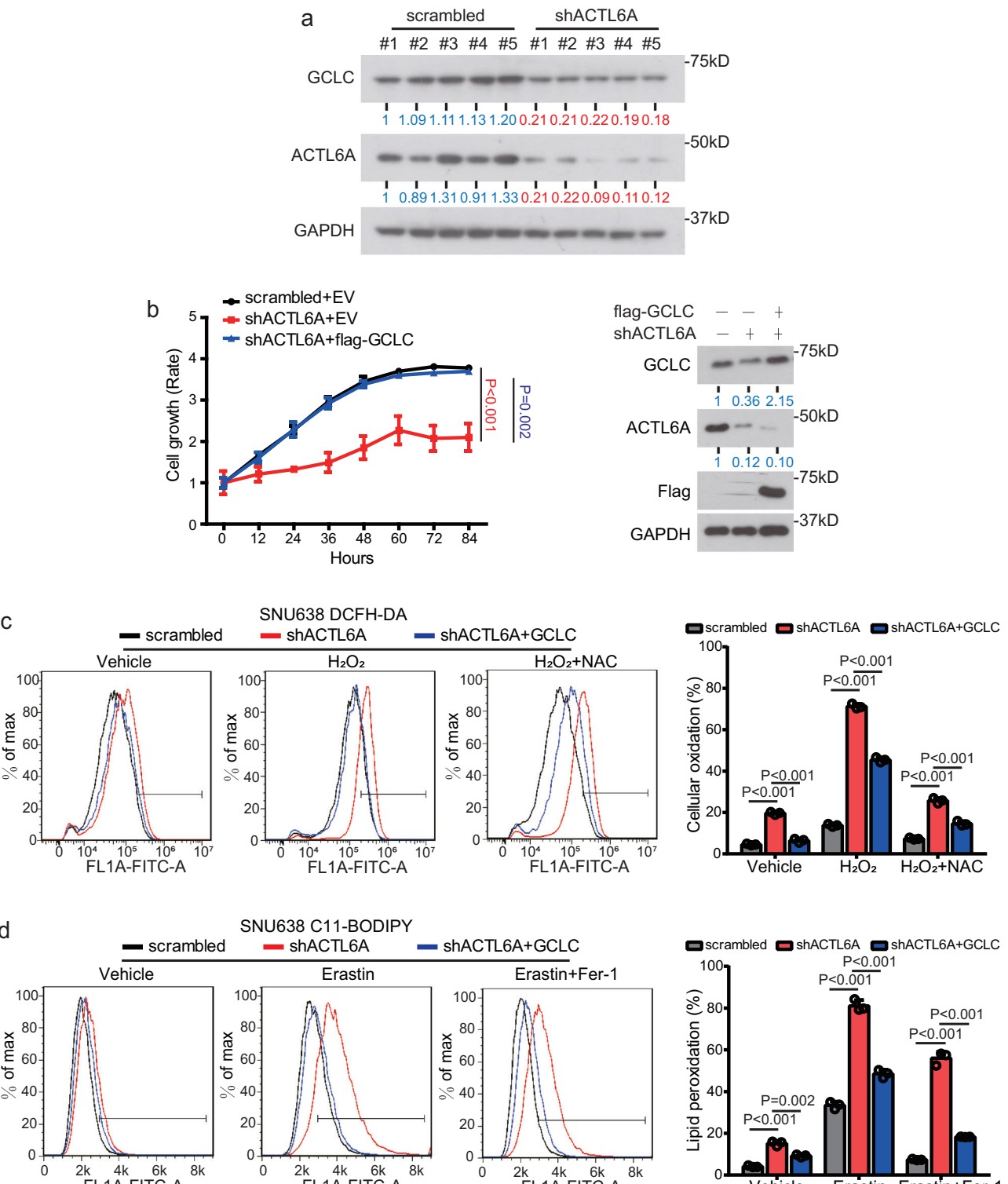

**Fig. 5 | ACTL6A inhibits GC cell ferroptosis via regulating GCLC.**
**a** Immunoblotting analysis of ACTL6A and GCLC protein levels in xenograft tissues derived from ACTL6A-knockdown SNU638 cells in mice. **b** Relative cell growth rate of SNU638 cells treated with ACTL6A shRNA or scrambled shRNA, and transfected with flag-GCLC or pcDNA3.1 vector. The Data are presented as the means ± SEM, $n = 3$ biologically independent experiments. Immunoblotting analysis of the indicated proteins is shown. **c** Relative DCFH-DA fluorescence measured by flow cytometry of cells treated with ACTL6A shRNA or scrambled shRNA, and cultured

with either $H_2O_2$ (50 μM) or both of $H_2O_2$ (50 μM) and NAC (100 μM) for 24 h. The data are presented as the means ± SD, $n = 3$ biologically independent experiments. **d** Relative C11-BODIPY fluorescence measured by flow cytometry of cells treated with ACTL6A shRNA or scrambled shRNA, and cultured with either erastin (10 μM), or both of erastin (10 μM) and Fer-1 (1 μM) for 24 h. The data are presented as the means ± SD, $n = 3$ biologically independent experiments. *P* values were determined by two-way ANOVA followed by Tukey test for panels **b**, and unpaired two-tailed T test for panels **c**, **d**.

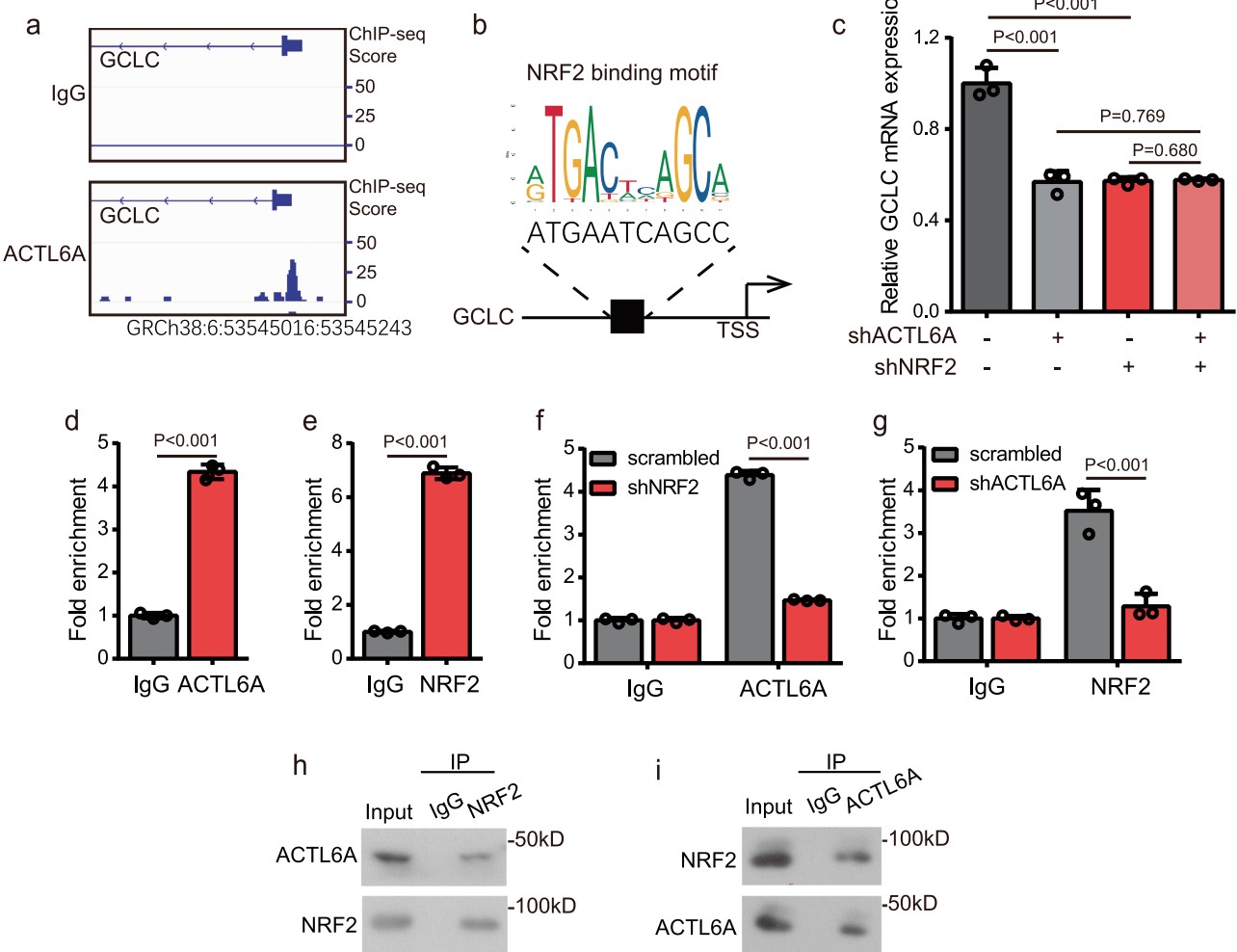

**Fig. 6 | ACTL6A transcriptionally regulates GCLC dependent on NRF2. a** Trace from ACTL6A ChIP-seq in SNU638 cells showing a binding peak upstream of the transcriptional start of GCLC. IgG was used as a control. **b** Predicting binding site of NRF2 in the peak from ChIP-seq result. NRF2-binding motif is from the website: http://jaspar.genereg.net/. **c** mRNA expression levels of GCLC in SNU638 cells treated with ACTL6A shRNA or scrambled shRNA, together with or without shNRF2. Data are presented as the means ± SD, $n = 3$ biologically independent experiments. **d, e** ChIP assay was performed in SNU638 cells using anti-ACTL6A (**d**) or anti-NRF2 (**e**) antibodies, followed by RT-qPCR with primers recognizing the predicting binding site of NRF2 in the transcriptional start of GCLC. The fold expression of ChIP-enriched mRNAs relative to the input was calculated, presented as the means ± SD, $n = 3$ biologically independent experiments. IgG was used as a control. **f** ChIP assay was performed in SNU638 cells treated with NRF2 shRNA or scrambled shRNA using anti-ACTL6A or anti-IgG antibodies, followed by RT-qPCR with

primers recognizing the predicting binding site of NRF2 in the transcriptional start of GCLC. The fold expression of ChIP-enriched mRNAs relative to the input was calculated. The data are presented as the means ± SD, $n = 3$ biologically independent experiments. **g** ChIP assay was performed in SNU638 cells treated with ACTL6A shRNA or scrambled shRNA using anti-NRF2 or anti-IgG antibodies, followed by RT-qPCR with primers recognizing the predicting binding site of NRF2 in the transcriptional start of GCLC. The fold expression of ChIP-enriched mRNAs relative to the input was calculated. The data are presented as the means ± SD, $n = 3$ biologically independent experiments. **h** Immunoblot analysis of the ACTL6A and NRF2 from anti-NRF2 immunoprecipitates (IP) obtained from SNU638 cells. IgG serves as a control. **i.** Immunoblot analysis of the NRF2 and ACTL6A from anti-ACTL6A immunoprecipitates (IP) obtained from SNU638 cells. IgG serves as a control. *P* values were determined by unpaired two-tailed T test for panels **c–g**.

To produce lentiviral particles, $1 \times 10^7$ HEK293T cells in a 55-cm² dish were co-transfected with 10 μg pLKO.1 shRNA construct, 5 μg psPAX2, and 5 μg pMD2G. The supernatant containing viral particles was harvested at 48 and 72 h after transfection, and was filtered through Millex-GP Filter Unit (0.45 μm pore size, Millipore). To infect cancer cells with lentivirus, cells were infected twice with culture medium containing 2 mL lentivirus, 200 μL FBS and 5 mg/ mL polybrene (Sigma) at 37 °C for 24 and 48 h. To increase the knockdown efficiency, infected cells were under several days of puromycin selection.

**Western blot analysis and immunoprecipitation**
Total cells lysates for immunoblotting or immunoprecipitation were lysed with buffer (0.1%Triton-100, 50 mM Tris-Cl, pH 7.5, 0.1% NP-40,

150 mM NaCl, 0.1 M EDTA) supplemented with a cocktail of phosphate and proteinase inhibitors for 30 min at 4 °C. For immunoblot, samples were separated by SDS–PAGE. Antibodies specific for ACTL6A (Abcam, ab3882, 1:4000), GCLC (Abcam, ab190685, 1:4000), anti-NRF2 (Cell Signaling Technology, 12721 s, 1:2000), flag-tag (Sigma, F1804, 1:5000), and GAPDH (Proteintech, 10494-1-AP, 1:5000) were purchased from the indicated companies. For immunoprecipitation, cell lysates were prepared as before and rotated 1 mg proteins with antibody at 4 °C overnight, then immunoprecipitated by Protein A/G beads; or incubated cell lysates with M2 beads (flag-tag) at 4 °C overnight. The immunoprecipitates and input were subjected to western blot analysis. The uncropped and unprocessed scans are provided in the Source Data file.

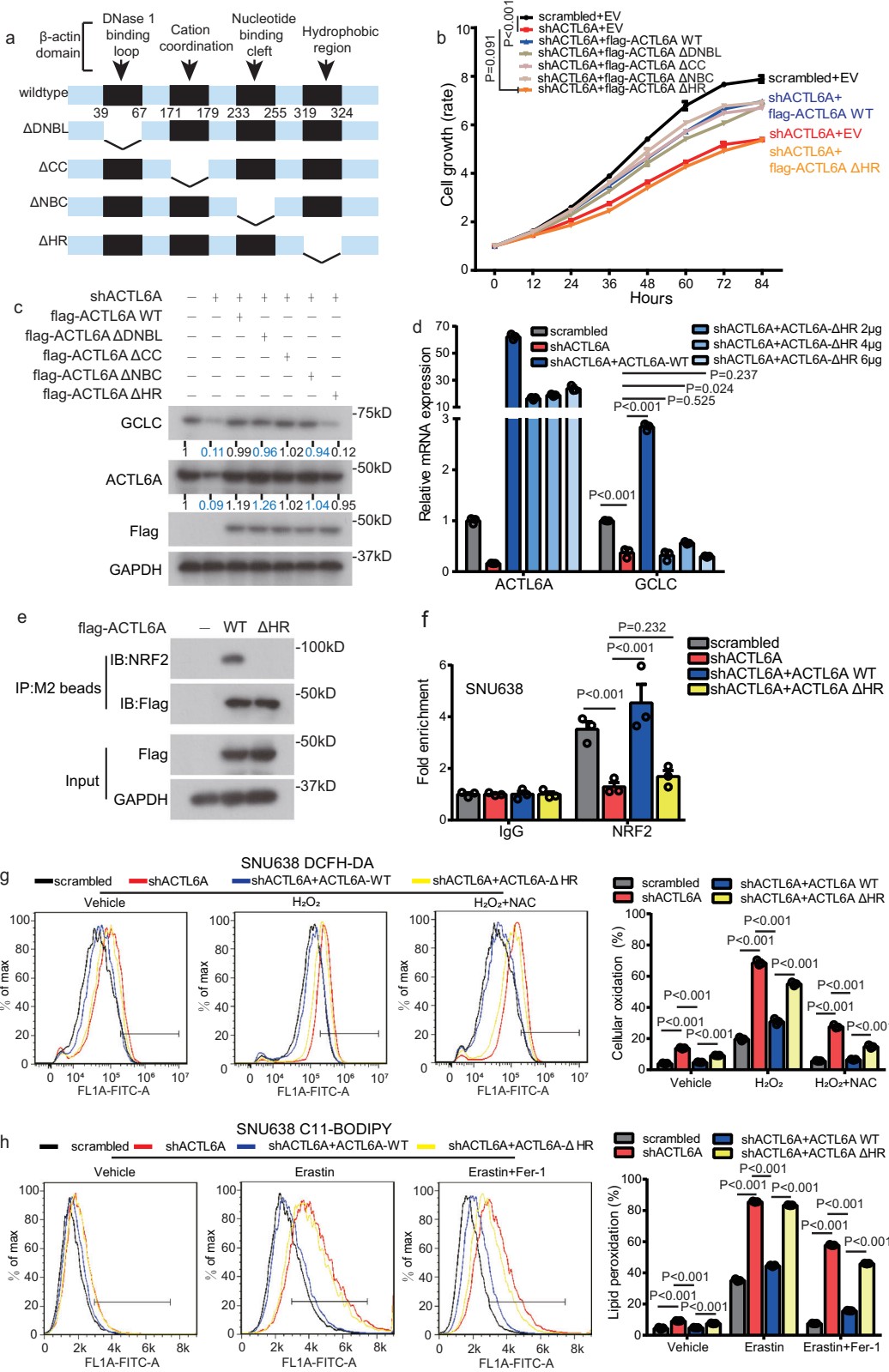

## Real-time qPCR

Total RNA was isolated with TRIZOL reagent (Invitrogen) and reverse transcription was performed by ReverTra Ace® qPCR RT Master Mix with gDNA Remover (TOYOBO), according to the manufacturer's protocol. Then the gene expression was quantified dependent on the LightCycler480 PCR system (Roche) by using 2×SYBR Green qPCR Master Mix (biotool, #B21203). All the target genes were normalized to GAPDH. Primer sequences are listed in Supplementary Table 2.

## Cell proliferation and viability

The IncuCyte system was used to examine cell proliferation. Cells (3 × 10⁵/well) were seeded in 6-well plates, then place the plates into the IncuCyte Live-Cell Analysis System (ESSEN BIOSCIENCE). After

**Fig. 7 | The HR domain of ACTL6A is essential in GCLC regulating and ferroptosis. a** A schematic drawing of ACTL6A domain deletion constructs. **b** Relative cell growth rate of SNU638 cells treated with ACTL6A shRNA or scrambled shRNA, and transfected with flag-ACTL6A-WT, constructs of domain deletion from (**a**) or pcDNA3.1 vector. The data are presented as the means ± SEM, $n = 3$ biologically independent experiments. **c** Immunoblotting analysis of the indicated proteins in cells from **b**. **d** mRNA levels of ACTL6A and GCLC in SNU638 cells treated with ACTL6A shRNA or scrambled shRNA, and transfected with ACTL6A-WT, increasing doses of ACTL6A-ΔHR or pcDNA3.1 vector. Data are presented as the means ± SD, $n = 3$ biologically independent experiments. **e** Immunoblot analysis of the indicated proteins from M2 beads immunoprecipitates (IP) and whole cell lysates (input) obtained from SNU638 cells transfected with ACTL6A-WT, ACTL6A-ΔHR, or pcDNA3.1 vector. **f** ChIP assay was performed in SNU638 cells treated with ACTL6A shRNA or scrambled shRNA, and transfected with ACTL6A-WT, ACTL6A-ΔHR, or pcDNA3.1 vector using anti-NRF2 or anti-IgG antibodies, followed by RT-qPCR with primers recognizing the predicting binding site of NRF2 in the transcriptional start of GCLC. The fold expression of ChIP-enriched mRNAs relative to the input was calculated, presented as the means ± SD, $n = 3$ biologically independent experiments. **g** Relative DCFH-DA fluorescence measured by flow cytometry of cells treated with scrambled or ACTL6A shRNA, and transfected with ACTL6A-WT or ACTL6A-ΔHR, and cultured with either $H_2O_2$ (50 μM) or both of $H_2O_2$ (50 μM) and NAC (100 μM) for 24 h. The Data are presented as the means ± SD, $n = 3$ biologically independent experiments. **h** Relative C11-BODIPY fluorescence measured by flow cytometry of cells treated with scrambled or ACTL6A shRNA, and transfected with ACTL6A-WT or ACTL6A-ΔHR, and cultured with either erastin (10 μM), or both of erastin (10 μM) and Fer-1 (1 μM) for 24 h. The data are presented as the means ± SD, $n = 3$ biologically independent experiments. $P$ values were determined by unpaired two-tailed T test for panels **d**, **f**–**h** and two-way ANOVA followed by Tukey test for panels **b**.

incubating at 37 °C, 5% $CO_2$ for 72–96 h, cell growth curve is obtained by analyzing cell confluence per well.

The cell counting kit-8 (CCK8) assay was used to examine cell viability. Cells (3000/well) were seeded in 96-well plates and were incubated for 24, 48, or 72 h. Subsequently, 10 μL of CCK8 (APExBIO) solution was added to each well and incubated for 3 h, then the absorbance value (OD) was measured at 450 nm. Data represent the means ± SD of three independent experiments.

### Wound-healing assay

Cells were cultured in six-well plates containing RPMI medium with 10% FBS. When cells grew to 90% confluence, they were starved for 24 h in serum-free medium. A sterile 10 μL pipette tip was used to create wounds, and areas of wound lines were observed and assessed by inverted microscope after 24 h. These experiments were performed in triplicate.

### Transwell migration assay

$2 \times 10^4$ cells in serum-free medium containing 0.1% bovine serum albumin were placed into the upper chamber of the insert and 500 μl complete medium were added into the bottom chamber. After 12 h of incubation cells were washed by PBS, the cells adhering to the lower membrane of the inserts was counted after staining with 0.1% crystal violet. The numbers of cells were counted under an inverted microscope.

### mRNA microarray and GC datasets availability

RNA samples from GC cell lines were isolated using the Trizol reagent, and then sent for sequence by Shanghai Biotechnology Corporation. The GC data sets were downloaded from the publicly available GEO databases and The Cancer Genome Atlas. Gene set enrichment analysis (GSEA) was performed by the JAVA program (https://www.gsea-msigdb.org/gsea/index.jsp) using KEGG v7.0. symbols gene set collection.

### Cell metabolism measurement

OCR and ECAR in real time were measured by the Seahorse Bioscience extracellular flux analyzer (XF24, Seahorse Bioscience). Briefly, 20,000–25,000 cells were seeded in specific 24-well plates designed for XF24 in 250 mL growth medium and incubated overnight. Prior to measurements, cells were washed with unbuffered medium once, immersed in 500 mL of unbuffered medium, and incubated in the absence of $CO_2$ for 1 h. The ECAR and OCR were then measured as recommended by Seahorse Bioscience.

Glutathione/glutathione disulfide (GSH/GSSG) ratio and Nicotinamide adenine dinucleotide phosphate NADPH/NADP+ ratio were measured using GSH/GSSG Assay Kit (S0053, Beyotime) and NADP/NADPH Quantification Kit (S0179, Beyotime) respectively. All these assays were performed according to the manufacturer's protocol.

### Intracellular ROS and lipid peroxidation level

Reactive oxygen species (ROS) were measured on the basis of the intracellular peroxide-dependent oxidation of DCFH-DA (S0033S, Beyotime) to form the fluorescent compound 2′,7′-dichlorofluorescein (DCF). Cells ($5 \times 10^5$ per well) were seeded in 6-well plates and cultured for 24 h. After washing twice with PBS, 2 μM DCFH-DA was added and cells were incubated for 30 min at 37 °C. The cells were washed twice with PBS, 400 μl of PBS were added to each well, and fluorescence intensity was determined with a flow cytometer (analyzed by FlowJo 7.6.1).

For lipid peroxidation level, Cells were treated with the fluorescent dye BODIPY™ 581/591 C11 (D3861, Thermofisher Scientific), which inserts into lipid membranes and allows for quantitative assessment of oxidized versus non-oxidized lipids by fluorescing green or red, respectively. Procedure of dying and detecting is the same as that of DCFH-DA and concentration of BODIPY 581/591 C11 is 5 μM.

For PDX tumors, frozen sections were established and stained with 10 μM DCFH-DA and 20 μM BODIPY 581/591 C11 for 30 min, and then washed twice by PBS. Fluorescence intensity of representative pictures was measured by the software image J.

### Metabolite pool and Isotopomer Labelling Analysis

For cell metabolic analysis, SNU638 cells were incubated with RPMI-1640 medium (glucose free or glutamine free), supplemented with 10% dialyzed FBS and 11 mM $[U^{13}C]$-glucose or 4 mM $[U^{13}C]$-glutamine for 24 h. The metabolites were extracted by 80% methanol with 2 μM myristic acid. Scraped the cell and centrifuged for 15 min at 12,000 g. The supernatant was transferred to LC–MS vials, and the pellet was lysed by 20 mM NaOH and protein levels were detected for normalization. A Dionex UltiMate 3000 506 LC System (Thermo Scientific) and a Q Exactive Orbitrap mass spectrometer (Thermo Scientific) operated in negative mode was used for targeted measurement. And calculation of the total carbon contribution ($^{13}C$ glucose/glutamine incorporation into metabolites) was corrected for naturally occurring isotopes (Siboyu Biotechnology Corporation, Guangzhou, China). Metabolite abundance was detected relative to the internal standard and normalized to the protein content.

For in vivo metabolic analysis, mice were intraperitoneally injected with a $^{15}N$-glutamine solution (40 mg/kg in PBS). After 2.5 h, PDX tumors were harvested, and weigh 20-30 mg for metabolite extraction by 80% methanol with 2 μM myristic acid. The later steps were the same as done in cells.

### ChIP-seq and ChIP assay

CUT&Tag assay was performed for chromatin immunoprecipitation/high-throughput sequencing (ChIP-seq). Briefly, $1 \times 10^5$ cells were washed gently with wash buffer (20 mM HEPES pH 7.5; 150 mM NaCl; 0.5 mM Spermidine; 1× Protease inhibitor cocktail). Add 10 μL

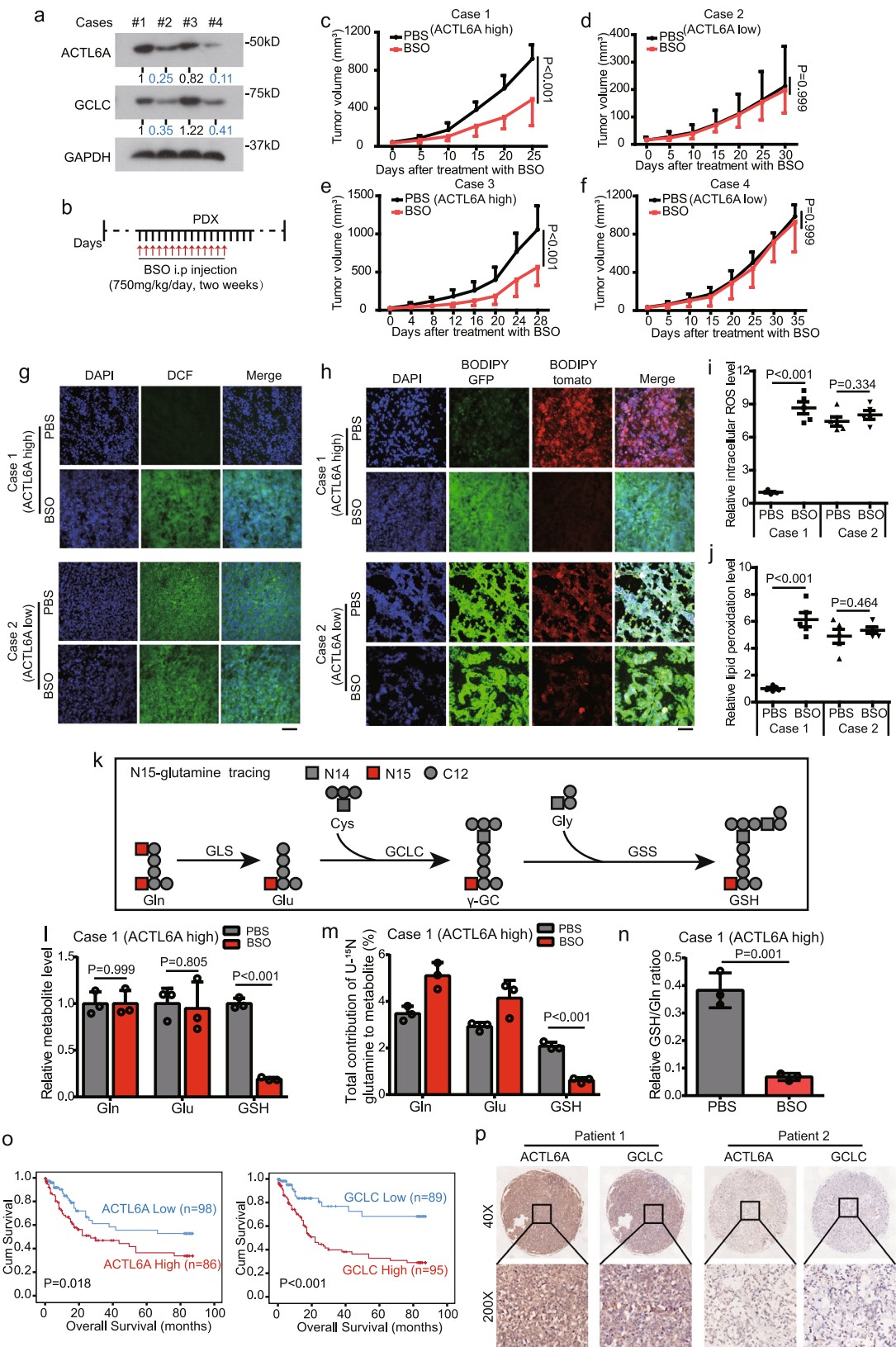

Concanavalin A coated magnetic beads (Bangs Laboratories) and incubate it at RT for 10 min. Remove supernatant and resuspended cells (bead-bound) with dig wash buffer (20 mM HEPES pH 7.5; 150 mM NaCl; 0.5 mM Spermidine; 1× Protease inhibitor cocktail; 0.05% Digitonin; 2 mM EDTA) and a primary antibody (anti-ACTL6A, Abcam, ab3882; anti-BRG1, Proteintech, 21634-1-AP; anti-NRF2, Cell Signaling

Technology, 12721 s) or IgG control antibody (normal rabbit IgG: Millipore cat.no. 12-370, normal mouse IgG: Millipore cat.no. 12-371) incubated on a roating platform overnight at 4 °C. The primary antibody was removed with a magnet stand. Secondary antibody (Anti-Rabbit IgG antibody, Goat monoclonal: Millipore AP132) was diluted 1:100 in dig wash buffer and cells were incubated at RT for 1 h. Wash

Fig. 8 | **Clinical relevance of the ACTL6A-GCLC-GSH metabolism axis in fer-roptosis of GC cells. a** Immunoblotting analysis of ACTL6A and GCLC protein levels in PDX tumors of 4 cases. **b**. The mice were treated with BSO (750 mg/kg/day) for two weeks. **c**–**f** Tumor volumes of PDX tumors, including case1 (**c**), case2 (**d**), case3 (**e**), and case4 (**f**). The data are presented as the means ± SD, $n = 6$ for each group in case 1 and 3, $n = 4$ for each group in case 2 and 4. **g**–**j** Representative fluorescent images of PDX frozen sections stained with DCFH-DA (**g**) and C11-BODIPY (**h**). Scale bars represent 200 μm. Relative ROS levels (**i**) and lipid perox-idation levels (**j**) were counted by image J and presented as bar graphs. The data are presented as the means ± SD, $n = 5$ for each group. BODIPY GFP represents lipid peroxidation; BODIPY tomato represents the original color of BODIPY. **k** Schematic metabolic map of [U-$^{15}$N]-labeled GSH de novo synthesis. Gln glutamine, Glu glutamate, Cys cysteine, Gly glycine, γ-GC γ-glutamyl-cysteine, GSH glutathione. **l**–**n**. Relative intracellular pool levels of glutamine, glutamate, and GSH in case1 (ACTL6A high) PDX tumors (**l**). Incorporation of nitrogen atoms from [U-$^{15}$N] glu-tamine into glutamine (gln), glutamate (glu), and GSH (**m**). Relative GSH/Gln ratios (**n**). Data are presented as the means ± SD, $n = 3$ for each group. **o** Kaplan–Meier curves of the overall survival of GC patients with different expression levels of ACTL6A and GCLC, and log-rank analysis was used to test for significance. **p** Representative images showing the correlation of ACTL6A and GCLC staining in human GC tissue microarray samples. Scale bars represent 50 μm. $P$ values were determined by two-way ANOVA followed by Tukey test for panels **c**–**f**, unpaired two-tailed T test for panels **i**, **j**, **l**–**n**, **h**, and the log-rank test for panel **o**.

cells with the magnet stand in dig wash buffer. Cells were incubated with a pA-Tn5 adapter complex in Dig-med buffer (0.01% Digitonin; 20 mM HEPES pH 7.5; 300 mM NaCl; 0.5 mM Spermidine; 1× Protease inhibitor cocktail). Cells were washed twice for 5 min in 1 mL Dig-med buffer. Then cells were resuspended in tag-mentation buffer (10 mM MgCl2 in Dig-med Buffer) and incubated at 37 °C for 1 h. DNA was purified using phenol-chloroform-isoamyl alcohol extraction and ethanol precipitation. This experiment was analyzed by Guangzhou Huayin Health Medical Group Co., Ltd.

ChIP assays were performed using the fast ChIP protocol with minor modifications. Briefly, cells were fixed for 15 min at RT with 1% formaldehyde. The sonicated chromatin was incubated with anti-bodies at 4 °C overnight with gentle rotation. The immunoprecipitates were mixed with protein A-agarose beads (sc2003, Santa Cruz) and rotated for 2 h at 4 °C. Chelex 100 resin (BioRad) was then added to the immunoprecipitates and input DNA samples, followed by incubation at 100 °C for 10 min to reverse crosslinking. After proteinase K treat-ment for 30 min, DNA was recovered and subjected to RT-PCR analysis using primers listed in Supplementary Table 3. The following anti-bodies were used for ChIP assay: anti-ACTL6A (Abcam, ab3882), anti-NRF2 (Cell Signaling Technology, 12721 s), and Rabbit IgG (ab37415) (Abcam, Cambridge, MA).

## Xenograft tumor model in nude mice

This experiment was approved by the Animal Ethical and Welfare Committee of the Sixth Affiliated Hospital of Sun Yat-sen University (Ethical code: IACUC-2021011501). $5 \times 10^6$ cells transduced with a scrambled shRNA or shACTL6A were inoculated subcutaneously into the hind-flanks of 6-week-old male BALB/ c-nu/nu mice. For the NAC rescue experiment, mice received intraperitoneal administra-tion of 200 mg/kg of body weight NAC (A105422, Aladdin) in phosphate-buffered saline (pH 7.4) every second day for two weeks. Tumor length and width were measured twice weekly, and the volume was calculated according to the formula (length ×width$^2$)/2. Sacrificed the mice when tumors reached approximately 15 mm at diameter, and dissected the tumors for weighting and other detections. All mice were housed under specific pathogen-free (SPF) conditions with 12 h light/12 h dark cycle at 21–24 °C, and humidity at 40–60%, and were fed with a standard chow diet at the Labora-tory Animal Center of the Sixth Affiliated Hospital, Sun Yat-sen University.

**Table 1 | Correlation of ACTL6A and GCLC staining in 184 human GC tissue microarray samples**

| Case number | | ACTL6A | | Sig. |
|---|---|---|---|---|
| ($n = 184$) | | High | Low | |
| GCLC | High | 62 | 33 | P < 0.001 |
| | Low | 24 | 65 | |

Chi-square tests were used to assess differences between groups.

## Patient-derived xenograft

The PDX experiment was approved by the Animal Ethical and Welfare Committee of the Sixth Affiliated Hospital, Sun Yat-sen University (Ethical code: IACUC-2021011501). Fresh patient samples were obtained from the Department of Surgery at the Sixth Affiliated Hos-pital of Sun Yat-sen University. All samples were collected with the patient's written informed consent. We detected ACTL6A protein expression of 48 GC samples, those higher than the upper quartile are defined as high ACTL6A group, and those lower than the lower quartile are defined as low ACTL6A group. We chose 2 cases in the high ACTL6A group and 2 cases in the low ACTL6A group for PDX experi-ment. GC samples Patient-derived tumor fragments (3-4mm$^3$) were surgically xenografted under the skin of 6-week-old male NCG mice. For BSO group, a week later, mice received intraperitoneal adminis-tration of 750 mg/kg of body weight BSO (B113387, Aladdin) in phosphate-buffered saline (pH 7.4) once a day for two weeks[25]. When tumors reached approximately 100 mm$^3$, mice were assigned ran-domly into one of two treatment groups. All mice were housed under specific pathogen-free (SPF) conditions with 12 h light/12 h dark cycle at 21–24 °C, and humidity at 40–60%, and were fed with a standard chow diet at the Laboratory Animal Center of the Sixth Affiliated Hospital, Sun Yat-sen University.

## Patient-derived organoid

Fresh GC PDX tumor tissues were cut into pieces, washed with ice-cold PBS, and then digested with EDTA. Following Matrigel polymerization for 10 min at 37 °C, advanced DMEM/F12 was supplemented with peni-cillin/streptomycin, 2 mM GlutaMAX, 10 mM HEPES, 1× B27, 1× N2 (Life Technologies), 1mM N-acetylcysteine (Sigma), and 10 nM gastrin I (Bio-gems). The following factors were used: 50 ng/mL recombinant EGF, 500 nM A83-01 (Biogems), 100 ng/ mL recombinant Noggin (Peprotech), 10 μM Y-27632 (Abmole), 500ng/mL R-spondin-1(Peprotech), 10 mM nicotinamide (Sigma), and 10 μM SB202190 (Sigma). When organoids were formed, we extracted original medium and used medium without NAC in the latter experiments. They were randomly separated into scrambled group and shACTL6A group via lentivirus transduction. For NAC or Fer-1 rescue group, organoids were treated with 100 μM NAC or 5 μM Fer-1. After treatment with shRNA lentivirus and NAC or Fer-1, we chose 8 PDOs at similar sizes to observe. These organoids were pictured every 3 days, and the sizes were measured with Image J. All samples were collected with the patients' written informed consent.

## Immunohistochemistry

Immunohistochemical staining was performed on paraffin sections prepared from xenograft or PDX tumor tissues. Briefly, sections were deparaffinized in xylene, hydrated in graded ethanol. To repair antigen, sections were microwaved for 15 min in EDTA, then immersed in 3% hydrogen peroxide to quench for endogenous peroxidase. Add block-ing buffer (goat serum) at RT for 1 h. Next, sections probed with primary antibody against ACTL6A (1:400, Abcam, ab3882), GCLC (1:200, Abcam, ab190685), Ki67 (1:400, Cell Signaling Technology, 9449) or 4-HNE

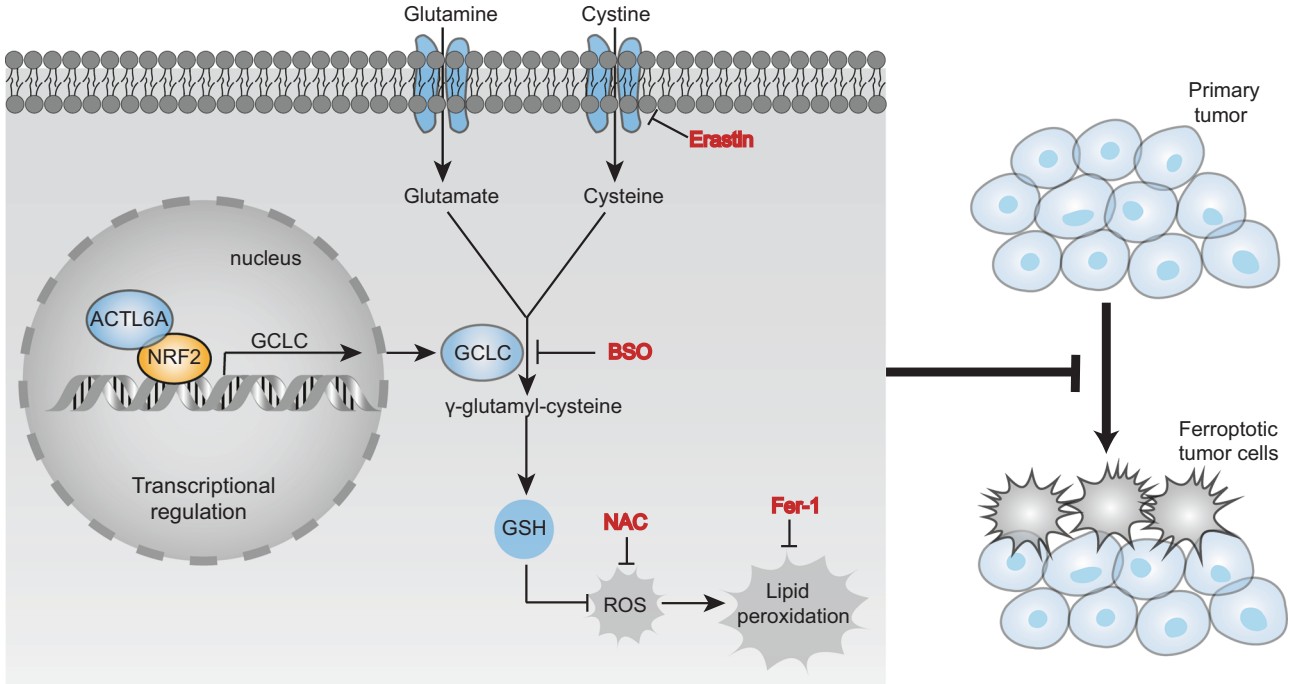

**Fig. 9 | Diagram of ACTL6A-mediated GSH synthesis and inhibits ferroptosis.** ACTL6A co-works with NRF2 on regulating GCLC in transcriptional level, and then promotes GSH synthesis and inhibits ferroptosis of GC cells.

(1:4000, Abcam, ab46545) at 4 °C overnight. Add secondary antibody for 20 min. Subsequently, immunostaining was visualized with diaminobenzidine and slices were counterstained with hematoxylin.

### Statistics and reproducibility

All statistical analyses were performed using SPSS 16.0. A nonpaired $t$ test was used to compare between groups. Two-way ANOVA was used to compare cell growth and tumor growth curves. Chi-square tests were used to assess differences in clinical variables between the GC cohorts. Cox proportional hazards regression analyses were used to assess the effect of clinical variables on patient survival. Kaplan–Meier survival analyses were used to compare survival among GC patients based on ACTL6A expression; the log-rank test was used to generate p values. Significance was defined as $P < 0.05$. All in vitro experiments were repeated three times with similar results.

### Reporting summary

Further information on research design is available in the Nature Portfolio Reporting Summary linked to this article.

## Data availability

The RNA array data generated in this study have been deposited in the Gene Expression Omnibus (GEO) database under accession code GSE203657. The anti-ACTL6A, anti-BRG1, and anti-NRF2 ChIP-seq data generated in this study have been deposited in the Gene Expression Omnibus (GEO) database under accession code GSE216350. Publicly available datasets reported in this paper are from the GEO databases (GSE13911, GSE27342, and GSE13861) and The Cancer Genome Atlas. Source data are provided with this paper.

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

## Acknowledgements

This research was supported by National Key R&D Program of China (2021YFF0702600), the National Natural Science Foundation of China (82222056 and 82111530099), Guangdong Special Young Talent Plan of Scientific and Technological Innovation (2019TQ05Y510), the Natural Science Foundation of Guangdong (2022A1515012316), the Guangdong International Joint Research Program (2020A0505100027), and National Key Clinical Discipline. We are grateful for Ling Fang from Instrumental Analysis & Research Center of Sun Yat-sen University for technical support.

## Author contributions

L.F., Z.Y. and S.Z. contributed to designing research studies. L.F., Z.Y., Y.Z., S.Z. and Y.X. contributed to analyzing data. L.F., Z.Y., S.Z., Y.Z., J.Z., L.X., M.M., J.F. and P.Z. contributed to conducting experiments. L.F., Z.Y. and Y.Z. contributed to writing the manuscript. L. K., M.H.L., and L.F. obtained funding.

## Competing interests

The authors declare no competing interests.
