## [Peer Review File · Nature Communications]

REVIEWER COMMENTS

Reviewer #1 (Remarks to the Author); expert in ACTL6A and epigenetics:

The study by Yang et al. entitled “ACTL6A Protects Gastric Cancer Cells 1 against Ferroptosis” reported a functional role of ACTL6A in gastric cancer. They found that ACTL6A was overexpressed in gastric cancer patients. Mechanistically, ACTL6A interacted with NRF2 to upregulate GCLC expression, thereby promoting GSH synthesis and prohibiting the ferroptosis of gastric cancer cells. They also found that ACTL6A-high cancer cells were significantly sensitive to BSO, which is an inhibitor of GCLC that serves as a ferroptosis inducer. This study may provide a potential therapeutic option by targeting the GSH metabolism in ACTL6A-high gastric cancer. However, the manuscript suffers from the fact that it's unclear how high vs. low ACTL6A will be defined and missing mechanistic links.

Major concerns:

1. Based on the data presented in Fig. 7, it is rather surprising that the response to BSO would be that different between Case 1 and Case 2 given the marginal differences in ACTL6A levels. Importantly, in order for the conclusion to be practical, how the high vs. low expression of ACTL6A would be defined in GC clinical samples?
2. Based on the analysis of the TCGA dataset, there is no difference in GCLC expression between normal and GC tumors. In addition, correlation between ACTL6A and GCLC should be validated in the TCGA GC dataset.
3. Would ACTL6A knockout show the same phenotype as ACTL6A knockdown?
4. Since ACTL6A is a subunit of the SWI/SNF complex, is the impaired GC cell proliferation caused by ACTL6A KD dependent on the SWI/SNF complex? Please include ChIP-seq for core subunits of the SWI/SNF complex for comparison. Along the same line, NRF2 ChIP-seq would be informative.
5. Fig 2a-b: The author stated that the glutathione metabolism was enriched in RNA-seq analysis (Fig 2a) and also can be enriched when cross-referencing the RNA array and two public datasets (Fig 2b). Is there a reason why the authors perform both RNA-seq and RNA array? In addition, does the GCLC expression change in the RNA-seq? It would also be informative to show some ACTL6A-perturbed genes that are involved in glutathione metabolism by heatmaps. Furthermore, the deposited GEO dataset GSE203657 was generated by RNA microarray rather than RNA-seq as stated.
6. Fig 2h-m: It would be informative to show whether NAC alters cellular oxidation with or without ACTL6A KD to consolidate the NAC rescued phenotypes in vitro and in vivo.
7. The author showed by the ChIP-seq analysis that ACTL6A bound to the promoter of GCLC. In addition to GCLC, are there any other genes involved in GSH metabolism that are associated with ACTL6A and whether their transcription can be changed upon ACTL6A KD (for example, GCLM)? A comprehensive and global analysis would be informative.

8. It would be informative to stain ferroptosis markers in xenografts treated with or without BSO. This can help consolidate the findings that the delayed tumor growth is due to ferroptosis. In addition, increased ROS by GCLC inhibition has been reported to cause apoptosis (ref 25). Would ACTL6A KD cause apoptosis?

9. Is BSO treatment tolerated in vivo? Please provide data on toxicity.

10. Published literature showed that NAC treatment suppresses the proliferation of cancer cells including gastric cancer cells. Given that ACTL6A is frequently upregulated in GC cells as stated, the authors should reconcile this discrepancy.

Minor concerns:

1. Please include positive controls for apoptosis and necroptosis inhibitor experiments to show that the assays worked at the intended concentration.

2. Fig. 1d should be “the log₂ T/N ACTL6A”?

3. Fig 1K-I, Fig 2m-n: Generally, the xenograft tumor volume and tumor weight should be comparable. This seems to be the case in Fig 1K-I, but not in Fig 2m-n.

4. Fig 6e: There is no Flag detection when immunoblotting the IP lysates of flag-ACTL6A ΔHR.

5. Typos: Line 77, “the that it roles in GC...”; Line 349 “C13 labeled glutamine”

6. Fig 5e: Please show the entire genomic loci of GCLC and also label the normalized read counts as Y-axis. The track of IgG control is also worth showing.

7. Figure missing: “similarly, ACTL6A KD restored the decrease in GCLC caused by NRF2 KD”, and the statement is not consistent with the model.

8. Minor: the manuscript is poorly written and quite hard to follow.

Reviewer #2 (Remarks to the Author); expert in ferroptosis:

Review of manuscript NCOMMS-22-18381-T, "ACTL6A Protects Gastric Cancer Cells against Ferroptosis through Induction of Glutathione Synthesis ", by Yang et al., et al.

In the present study, the authors study the impact of the loss of ACTL6A, an accessory component of multiple chromatin remodelling complexes, on tumour growth. The results demonstrate that loss of ACTL6A leads to a marked growth of tumours in vitro and in vivo. The authors subsequently analyse the transcriptional signature of ACTL6A knockdown cell lines and identified a potential role of glutathione metabolism and ferroptosis induction as the underlying cause. Following up on these observations the authors could link these effects to the binding of ACTL6A to Nuclear factor (erythroid-derived 2)-like 2

(NFE2L2, a.k.a NRF2), a transcription factor regulating the oxidative stress response. The results are generally interesting and could have implications for a better understanding of the process of ferroptosis and also provide novel drug targets for GC.

Having said this, I have a few remarks the authors might consider addressing :

- The in vivo experiments showed a marked decrease of tumours expressing the shACTL6A, given the reported essentiality of ACTL6A it would be interesting to know what are the levels of ACTL6A in the “escapers”. Can they proliferate in the absence of ACTL6A or you are just selecting for cells that have low knockdown.
- The authors have mostly used one cell line (SNU638) to address the mechanism proposed here. I would encourage them to validate some of their findings in additional GC cell lines in order to better establish the impact of their findings. Moreover, this would also encourage these analyses in non-GC GC cell lines to delineate if this mechanism is widespread or lineage-specific.
- ACTL6A is reported as a common essential gene in multiple genome-wide CRISPR screens (www.depmap.org) and yet little or no co-essentiality is shared with known regulators of ferroptosis. Therefore, it's not clear if the cell death or growth impairment observed is only due to ferroptosis. Can the complete loss of ACTL6A (knockout) be rescued by liproxstatin treatment or GCLC overexpression?
- The authors should also better characterize the ACTL6A cell in regard to other ferroptosis regulators, including GPX4, FSP1, DHODH, SLC7A11, and ACSL4 – for example.
- Along similar lines, challenging ACTL6A with ferroptosis-inducing agents that inhibit GPX4, FSP1 or DHODH could be informative.

Minor comments –

In Figure 3h, it's interesting to note that the organoids have remarkable morphological differences, especially the Fer1 treated ones. Maybe the author care to comment on this.

In Figure 6c it would encourage the authors to write the exact deletions

In Figure 6e it appears that the Flag and NRF2 blots are swapped.

Reviewer #3 (Remarks to the Author); expert in gastric cancer:

In the manuscript entitled “ACTL6A Protects Gastric Cancer Cells against Ferroptosis through Induction of Glutathione Synthesis”, the authors applied a series of studies to investigate the correlation between

ACTL6A and ferroptosis by regulating GSH metabolism; They find ferroptosis was inhibited by ACTL6A by increasing GSH level, highlights the importance of ACTL6A in gastric cancer cell metabolism and tumorigenesis. The conclusion is potentially interesting, I have several concerns that limited my passion.

Major comments

1. The authors make conclusions that ACTL6A regulated gastric cancer tumorigenesis, however, no GC tumorigenesis data were shown, and ACTL6A null mouse-based spontaneous tumorigenesis experiments should be applied. Xenograft of cell line/PDOs in nude mice is not sufficient to support the authors' conclusion.
2. The size of the GC PDOs seems too small, only ranging from 20 μ m to 80 μ m, which is much smaller than published GC PDOs data. Had the authors applied further experiments to identify whether they are truly cancer PDOs?
3. The quantification and statistical methods of the PDOs should be detailed in the methods, for example, what does each dot in statistical plots indicated? One single PDO, one repeated well, or one PDO case? Moreover, all experiments must be repeated at least 3 times.
4. In methods, the authors claimed basic PDO culture medium already contains 1mM NAC, however, data in Fig2h-I showed a dramatic difference when adding 100 μ m additional NAC, the accuracy of the data is questionable.
5. SNU638 cell is the only gastric cancer cell line used in this paper. Making conclusions in only one cell line is dangerous.
6. The ACTL6A seems nuclear expression in Fig 3i but cell plasma expression in Fig 7q. The authors should explain the deviation between them.

Minor comments

1. Measured the grayscale in all Western blot bands.
2. X axis in Fig1d is missed.

Reviewer #1 (Comments to the Author):

The study by Yang et al. entitled "ACTL6A Protects Gastric Cancer Cells 1 against Ferroptosis" reported a functional role of ACTL6A in gastric cancer. They found that ACTL6A was overexpressed in gastric cancer patients. Mechanistically, ACTL6A interacted with NRF2 to upregulate GCLC expression, thereby promoting GSH synthesis and prohibiting the ferroptosis of gastric cancer cells. They also found that ACTL6A-high cancer cells were significantly sensitive to BSO, which is an inhibitor of GCLC that serves as a ferroptosis inducer. This study may provide a potential therapeutic option by targeting the GSH metabolism in ACTL6A-high gastric cancer. However, the manuscript suffers from the fact that it' s unclear how high vs. low ACTL6A will be defined and missing mechanistic links.

Major concerns:

1. Based on the data presented in Fig. 7, it is rather surprising that the response to BSO would be that different between Case 1 and Case 2 given the marginal differences in ACTL6A levels. Importantly, in order for the conclusion to be practical, how the high vs. low expression of ACTL6A would be defined in GC clinical samples?

→A: We appreciate the reviewer's comments. Because higher ACTL6A in tumor than in normal tissue indicates that ACTL6A play an important role in the tumor. When conducting PDX experiments, we chose GC clinical samples those expressed higher ACTL6A than the adjacent normal tissue (at least two times) as ACTL6A-high samples, and those expressed nearly the same ACTL6A as ACTL6A-low samples (Rebuttal Fig. 1). ACTL6A expressions in case1 and 3 (ACTL6A-high) is much higher than case 2 and 4 (Fig. 7a). Compared with control group, BSO can inhibit about half of the tumor growth in ACTL6A-high samples. There is no significant difference in ACTL6A-low samples (Fig. 7a-f, Supplementary Fig. 7a).

Rebuttal Figure 1. Immunoblot analysis of ACTL6A protein levels in PDX tumors (T) and adjacent normal tissue (N) of 4 cases.

Figure 7a. Immunoblotting analysis of ACTL6A and GCLC protein levels in PDX tumors of 4 cases.

2. Based on the analysis of the TCGA dataset, there is no difference in GCLC expression between normal and GC tumors. In addition, correlation between ACTL6A and GCLC should be validated in the TCGA GC dataset.

→We appreciate the reviewer's comments. We searched the website UALCAN (<http://ualcan.path.uab.edu/>) and found that expression of GCLC was higher in GC tumors than in normal tissue (Rebuttal Fig. 2a). Besides, GCLC was found to be positively correlated to ACTL6A in TCGA samples (Rebuttal Fig. 2b).¹ And we also validated it from 21 fresh GC samples (T/N) (Rebuttal Fig. 2c, d).

Rebuttal Figure 2. a. Expression of GCLC in GC tumor tissues and normal tissues. Data are from <http://ualcan.path.uab.edu/>, $p < 0.001$. b. Correlation of ACTL6A and GCLC expression (log2) in TCGA GC samples. c. Waterfall plot of the relative GCLC mRNA levels from 21 paired samples of GC and normal tissue measured using qRT-PCR. The log2 T/N ACTL6A mRNA levels are presented. d. Correlation of ACTL6A and GCLC expression (log2 [T/N]) in 21 clinical GC samples.

3. Would ACTL6A knockout show the same phenotype as ACTL6A knockdown?
 → We appreciate the reviewer's comments. We built up ACTL6A knockout (KO) cells by CRISPR-cas9 gene editing. Then we found that ACTL6A KO decreased GCLC expression in protein level and mRNA level (Rebuttal Fig. 3a, b). Besides, ACTL6A KO cells expressed higher PTGS2, a ferroptosis marker (Rebuttal Fig. 3b). ACTL6A KO decreased GSH/GSSG ratio (Rebuttal Fig. 3c). ACTL6A KO inhibited cell growth significantly, which could be rescued by liproxstatin-1 treatment and GCLC overexpression (Rebuttal Fig. 3d, e).

Rebuttal Figure 3. a. Immunoblot analysis of ACTL6A and GCLC protein levels in parental SNU638 cells (sgRNA Con) and ACTL6A knockout SNU638 cells (sgACTL6A-1 and sgACTL6A-2). b. mRNA expression levels of ACTL6A, GCLC and PTGS2 in parental SNU638 cells (sgRNA Con) and ACTL6A knockout SNU638 cells (sgACTL6A-1 and sgACTL6A-2). The data are presented as the means \pm SD, $***P < 0.001$. c. Measurement of relative GSH/GSSG ratio in parental SNU638 cells (sgRNA Con) and ACTL6A knockout SNU638 cells (sgACTL6A-1 and sgACTL6A-2). The data are presented as the means \pm SD, $**P < 0.01$. d. Relative cell growth rate of parental SNU638 cells and ACTL6A knockout SNU638 cells, and treated with or without liproxstatin-1 for the indicated time points. The data are presented as the means \pm SEM, $***P < 0.001$. e. Relative cell growth rate of parental SNU638 cells and

ACTL6A knockout SNU638 cells, and transfected with or without flag-GCLC for the indicated time points. The data are presented as the means \pm SEM, *** P <0.001.

4. Since ACTL6A is a subunit of the SWI/SNF complex, is the impaired GC cell proliferation caused by ACTL6A KD dependent on the SWI/SNF complex? Please include ChIP-seq for core subunits of the SWI/SNF complex for comparison. Along the same line, NRF2 ChIP-seq would be informative.

→We appreciate the reviewer's comments. We have conducted an experiment of anti-BRG1 (the catalytic subunit of the SWI/SNF chromatin-remodeling complex) ChIP-seq. The results showed that BRG1 could bind to the TSS of GCLC, and the peak (GRCh38:6:53544791:53545250) was overlapped with that of anti-ACTL6A ChIP-seq (GRCh38:6:53545016:53545243) (Rebuttal Fig. 4). This result is also showed in Supplementary Fig. 5p. Therefore, the function of ACTL6A in GC is dependent on the SWI/SNF complex.

Besides, we have done the NRF2 ChIP-seq which confirmed that NRF2 could also bind to the TSS of GCLC (Rebuttal Fig. 4).

Rebuttal Figure 4. The entire genomic loci of GCLC in ACTL6A, BRG1 and NRF2 ChIP-seq results. IgG was used as a control.

5. Fig 2a-b: The author stated that the glutathione metabolism was enriched in RNA-seq analysis (Fig 2a) and also can be enriched when cross-referencing the RNA array and two public datasets (Fig 2b). Is there a reason why the authors perform both RNA-seq and RNA array? In addition, does the GCLC expression change in the RNA-seq? It would also be

informative to show some ACTL6A-perturbed genes that are involved in glutathione metabolism by heatmaps. Furthermore, the deposited GEO dataset GSE203657 was generated by RNA microarray rather than RNA-seq as stated.

→ We appreciate the reviewer's comments. We are sorry that we miswrote the sentences. All of the "RNA-sequence" mentioned in the paper should be RNA array. And we have rephrased the sentences. (Line 151). Genes involved in glutathione metabolism from RNA array results were shown by heatmaps (Rebuttal Fig. 5), which we have shown in Supplementary Fig. 2a.

Rebuttal Figure 5. Heatmap of GSH metabolism related genes in RNA array results. The color scale indicates fold change of log₂ signal intensities of indicated genes.

6. Fig 2h-m: It would be informative to show whether NAC alters cellular oxidation with or without ACTL6A KD to consolidate the NAC rescued phenotypes in vitro and in vivo.

→ Thanks a lot for the reviewer's comments. We have detected cellular ROS levels by DCFH-DA staining and showed that NAC could rescued the cellular oxidation levels with or without ACTL6A KD (Rebuttal Fig. 6a-b). Knocking down ACTL6A expression with shACTL6A significantly increases cellular oxidation, and NAC rescued this effect in xenograft tumors (Rebuttal Fig. 6c).

Therefore, NAC rescued phenotypes which were caused by ACTL6A knockdown in vitro and in vivo.

Rebuttal Figure 6. a-b. DCFH-DA fluorescence measured by flow cytometry of GC cells (**a**: SNU638, **b**: SNU216) treated with ACTL6A shRNA or scrambled shRNA and cultured with or without 100 μ M NAC for 24h. The data are presented as the means \pm SD, ** P <0.01, *** P <0.001. **c.** Representative fluorescent images of xenograft tumors frozen sections stained with DCFH-DA. Scale bars represent 200 μ m.

7. The author showed by the ChIP-seq analysis that ACTL6A bound to the promoter of GCLC. In addition to GCLC, are there any other genes involved in GSH metabolism that are associated with ACTL6A and whether their

transcription can be changed upon ACTL6A KD (for example, GCLM)? A comprehensive and global analysis would be informative.

→We appreciate the reviewer's comments. From ChIP-seq results, we found that in addition to GCLC, ACTL6A might bind to some other genes involved in GSH metabolism: GCLM, SLC7A11 and GSS (Rebuttal Fig. 7a). There is no peak in the entire genomic loci of GPX4 and GSR (Rebuttal Fig. 7a). Moreover, ACTL6A KD could decrease GCLC, SLC7A11 and GSS, and GCLC was the most significantly decreased gene (Rebuttal Fig. 7b). And ACTL6A KD or overexpression had no impact on GCLM expression (Supplementary Fig. 5a-b). Therefore, we believe that ACTL6A might promote GSH synthesis via regulating many GSH metabolism related genes, but GCLC is the most valuable target.

Rebuttal Figure 7. a. Trace from ACTL6A ChIP-seq in SNU638 cells, the entire genomic loci of GCLC, GCLM, GPX4, SLC7A11, GSS and GSR were shown. **b.** mRNA expression levels of GCLC, SLC7A11 and GSS in GC cells treated with ACTL6A shRNA or scrambled shRNA.

Supplementary figure 5. a. mRNA expression levels of ACTL6A, GCLC and GCLM in SNU638 cells treated with ACTL6A shRNA or scrambled shRNA. Data are representative of three repeated experiments, *** $P < 0.001$, ns=no significance. **b.** mRNA expression levels of ACTL6A, GCLC and GCLM in SNU668 cells transfected with flag-ACTL6A or pcDNA3.1

vector. Data are representative of three repeated experiments, $***P<0.001$, ns =no significance.

8. It would be informative to stain ferroptosis markers in xenografts treated with or without BSO. This can help consolidate the findings that the delayed tumor growth is due to ferroptosis. In addition, increased ROS by GCLC inhibition has been reported to cause apoptosis (ref 25). Would ACTL6A KD cause apoptosis?

→We appreciate the reviewer's comments. In our primary submission, we have showed that BSO treatment could cause ferroptosis in PDXs by DCFH-DA and C11-BODIPY staining (Fig. 7g-j), and IHC staining for 4-HNE, a ferroptosis marker. (Supplementary Fig. 7b).

Increased ROS by GCLC inhibition has been reported to cause apoptosis (ref 25). We then found that ACTL6A KD may cause apoptosis by IHC staining xenografts for cleaved-caspase3, an apoptosis marker (Rebuttal Fig. 8). And our data showed that apoptosis inhibitor ZVAD-FMK could not rescue the decreased cell viability caused by ACTL6A KD (Fig 3a). Therefore, ACTL6A KD might cause apoptosis, but ferroptosis play a more important role in this process.

Rebuttal Figure 8. IHC staining for cleaved-caspase3 in xenograft tissues derived from ACTL6A-knockdown SNU638 cells in mice. Scale bars represent 50 μ m.

Figure 3. a. Viability of SNU638 cells treated with ACTL6A shRNA or scrambled shRNA after 24 hours cultured with or without 1 μ M Fer-1, 2 μ M necrostatin-1, 5 μ M Z-VAD-FMK. The data are presented as the means \pm SD. ** P <0.01, ns =no significance.

9. Is BSO treatment tolerated in vivo? Please provide data on toxicity.

→We appreciate the reviewer's comments. To study the toleration of BSO, NCG mice received intraperitoneal administration of 750 mg/kg of body weight BSO once a day for 4 weeks, then we collected organs (heart, liver, spleen, lung, kidney, stomach) from mice. We did the HE staining and showed that BSO treatment had no significant toxic effect in vivo (Rebuttal Fig. 9).

Rebuttal Figure 9. HE staining of organs (heart, liver, spleen, lung, kidney, stomach) from mice treated with BSO or PBS. Scale bar represents 100 μ m.

10. Published literature showed that NAC treatment suppresses the proliferation of cancer cells including gastric cancer cells. Given that ACTL6A is frequently upregulated in GC cells as stated, the authors should reconcile this discrepancy.

→We appreciate the reviewer's comments. Recent studies suggested NAC supplementation can promote cancer cell proliferation of multiple types of cancer, including gastric cancer, lung cancer and colorectal cancer²⁻⁴, which was consistent with our findings. Our findings indicated that ACTL6A, which was frequently upregulated in GC cells, decreased cellular ROS levels by promoting GSH synthesis, thereby promoting GC cell proliferation. Therefore, when ACTL6A was knocked down, cellular ROS levels were increased, then NAC could rescue the cell proliferation for its function as an antioxidant.

Minor concerns:

1. Please include positive controls for apoptosis and necroptosis inhibitor experiments to show that the assays worked at the intended concentration.

→We appreciate the reviewer's suggestions. To assured that the assays worked at the intended concentration, we induced apoptosis in GC cells with 40μM etoposide and found that the decreased cell viability could be rescued by 5μM ZVAD-FMK (Rebuttal Fig. 10a). Besides, 2μM necrostatin-1 could rescued the decreased cell viability induced by Necroptosis Inducer Kit by TSZ (TNF-α, SM-164 and ZVAD-FMK) (Rebuttal Fig. 10b). Therefore, the intended concentration of ZVAD-FMK and necrostatin-1 could work, which indicated that apoptosis and necroptosis might be induced by ACTL6A KD in GC cells, but they did not play an important role.

Rebuttal Figure 10. a. Viability of SNU638 cells treated with 40μM etoposide, and rescued by 5μM Z-VAD-FMK. The data are presented as the means ± SD. *** $P < 0.001$. **b.** Viability of SNU638 cells treated with Necroptosis Inducer Kit with TSZ (TNF-α, SM-164 and ZVAD-FMK), and rescued by 2μM necrostatin-1. The data are presented as the means ± SD. *** $P < 0.001$.

2. Fig. 1d should be “the log₂ T/N ACTL6A” ?

→We appreciate the reviewer's comments. We have corrected the sentences in Figure legends. (Line 855)

3. Fig 1k-1, Fig 2m-n: Generally, the xenograft tumor volume and tumor

weight should be comparable. This seems to be the case in Fig 1K-I, but not in Fig 2m-n.

→We appreciate the reviewer's comments. The shapes of some xenograft tumors were sometimes irregular when in vivo, which might account for the fact that the xenograft tumor volume and tumor weight were not totally match in Fig 2m-n. And our data were based on real measurement.

4. Fig 6e: There is no Flag detection when immunoblotting the IP lysates of flag-*ACTL6A* ΔHR.

→We appreciate the reviewer's comments. We are sorry that we labeled "Flag" and "NRF2" at a reverse place. We have corrected it in Figure 6e.

5. Typos: Line 77, "the that it roles in GC..." ; Line 349 "C13 labeled glutamine"

→We appreciate the reviewer's comments. We have rephrased the sentences: **Line 76-78**, "Although *ACTL6A* has been characterized as an oncogene in many human cancers, its role in GC is unknown, and knowledge of the underlying mechanisms is limited." **Line 360**, "and the levels of [¹⁵N]-labeled glutamine and glutamate were thus increased."

6. Fig 5e: Please show the entire genomic loci of *GCLC* and also label the normalized read counts as Y-axis. The track of IgG control is also worth showing.

→We appreciate the reviewer's comments. We have showed the entire genomic loci of *GCLC* (**Rebuttal Fig. 11**). We also labeled the normalized read counts as Y-axis and showed the track of IgG control (**Fig. 5e**).

Rebuttal Figure 11. The entire genomic loci of GCLC in ACTL6A ChIP-seq results. IgG was used as a control.

Figure 5. e. Trace from ACTL6A ChIP-seq in SNU638 cells showing a binding peak upstream of the transcriptional start of GCLC. IgG was used as a control.

7. Figure missing: “similarly, ACTL6A KD restored the decrease in GCLC caused by NRF2 KD” , and the statement is not consistent with the model.

→We appreciate the reviewer’s comments. We might not state it clearly, we meant to showed that NRF2 promoted the transcription of GCLC with the help of ACTL6A, when ACTL6A was knocked down, NRF2 KD could not decrease GCLC expression. The data were showed in Figure 5g. We have rephrased the sentences. **Line 279-280**, “Similarly, when ACTL6A was knocked down, NRF2 KD could not decrease GCLC expression.”

8. Minor: the manuscript is poorly written and quite hard to follow.

→We appreciate the reviewer’s comments. We will try our best to modified it.

Reviewer #2 (Comments to the Author):

In the present study, the authors study the impact of the loss of ACTL6A, an accessory component of multiple chromatin remodelling complexes, on tumour growth. The results demonstrate that loss of ACTL6A leads to a marked growth of tumours in vitro and in vivo. The authors subsequently analyse the transcriptional signature of ACTL6A knockdown cell lines and identified a potential role of glutathione metabolism and ferroptosis induction as the underlying cause. Following up on these observations the authors could link these effects to the binding of ACTL6A to Nuclear factor (erythroid-derived 2)-like 2 (NFE2L2, a.k.a NRF2), a transcription factor regulating the oxidative stress response. The results are generally interesting and could have implications for a better understanding of the process of ferroptosis and also provide novel drug targets for GC.

1. The in vivo experiments showed a marked decrease of tumours expressing the shACTL6A, given the reported essentiality of ACTL6A it would be interesting to know what are the levels of ACTL6A in the “escapers”. Can they proliferate in the absence of ACTL6A or you are just selecting for cells that have low knockdown.

→We appreciate the reviewer’s comments. We planted GC cells to mice right after shRNA lentivirus infection without selecting. We did detect the levels of ACTL6A in xenograft tumors (Fig 5a), which showed that ACTL6A was quite low expressed. It means that ACTL6A KD could promote the process of GC cell ferroptosis, and some of them are still able to proliferate in a lower speed.

Figure 5a. Immunoblotting analysis of ACTL6A and GCLC protein levels in xenograft tissues derived from ACTL6A-knockdown SNU638 cells in mice.

2. The authors have mostly used one cell line (SNU638) to address the mechanism proposed here. I would encourage them to validate some of their findings in additional GC cell lines in order to better establish the impact of their findings.

→We appreciate the reviewer's suggestions. In our primary submission, we showed that ACTL6A promote GC cell proliferation in three cell lines (SNU638, SNU668 and SNU216) (Fig 1e-f, Supplementary Fig. 1b-c), and ACTL6A KD decreased GSH/GSSG ratio in SNU216 (Supplementary Fig. 2f). Besides, we also showed that NAC rescued ACTL6A-KD cell proliferation in SNU216 (Supplementary Fig. 2g). And in Supplementary Fig. 3a, we showed that Fer-1 rescued ACTL6A-KD cell viability in SNU216.

To further validate our findings, we conducted experiments in another GC cell line (SNU216). First of all, ACTL6A KD increased cellular ROS levels and lipid peroxidation levels of SNU216 (Rebuttal Fig. 12a-b). ACTL6A regulates GCLC in transcriptional level via NRF2 (Rebuttal Fig. 13a-j). The HR domain of ACTL6A is essential in GCLC regulating and ferroptosis (Rebuttal Fig. 14a-f). The results indicated that our findings could be repeated in another GC cell line. These data have been shown in Supplementary Information.

Rebuttal Figure 12. a. DCFH-DA fluorescence intensity measured by flow cytometry, SNU216 cells are treated with ACTL6A shRNA and scrambled shRNA and cultured with or without 50 μ M H₂O₂ for 24 hours. Data are representative of three repeated experiments, *** P <0.001. **b.** C11-BODIPY fluorescence measured by flow cytometry of SNU216 cells treated with ACTL6A shRNA or scrambled shRNA and cultured with either erastin (10 μ M), Fer-1 (1 μ M) or both for 24 hours. The percentages of lipid peroxidation are presented as the means \pm SD, *** P <0.001.

Rebuttal Figure 13. a. Relative cell growth rate of SNU216 cells treated with ACTL6A shRNA or scrambled shRNA, and transfected with flag-GCLC or pcDNA3.1 vector for the indicated time points. The Data are presented as the means \pm SEM, * P <0.05, ** P <0.01. **b.** Immunoblot analysis of the NRF2, GCLC and ACTL6A protein levels in SNU216 cells treated with ACTL6A shRNA or scrambled shRNA. **c.** Immunoblot analysis of the ACTL6A and NRF2 from anti-NRF2 immunoprecipitates (IP) obtained from SNU216 cells. Immunoglobulin G (IgG) serves as a control. **d.** Immunoblot analysis of the NRF2 and ACTL6A from anti-ACTL6A immunoprecipitates (IP) obtained from SNU216 cells. Immunoglobulin G (IgG) serves as a control. **e-f.** Relative DCFH-DA (**e**) and C11-BODIPY (**f**) fluorescence measured by flow cytometry of SNU216 cells with indicated treatment. The data are presented as the means \pm SD, ** P <0.01, *** P <0.001. **g.** ChIP assay was performed in SNU216 cells using anti-ACTL6A or anti-IgG antibodies, followed by RT-qPCR with primers recognizing the predicting binding

site of NRF2 in the transcriptional start of GCLC. The fold expression of ChIP-enriched mRNAs relative to the input was calculated. The data are presented as the means \pm SD, *** P <0.001. **h.** ChIP assay was performed in SNU216 cells using anti-NRF2 or anti-IgG antibodies, followed by RT-qPCR with primers recognizing the predicting binding site of NRF2 in the transcriptional start of GCLC. The fold expression of ChIP-enriched mRNAs relative to the input was calculated. The data are presented as the means \pm SD, *** P <0.001. **i.** ChIP assay was performed in SNU216 cells treated with NRF2 shRNA or scrambled shRNA using anti-ACTL6A or anti-IgG antibodies, followed by RT-qPCR with primers recognizing the predicting binding site of NRF2 in the transcriptional start of GCLC. The fold expression of ChIP-enriched mRNAs relative to the input was calculated. The data are presented as the means \pm SD, *** P <0.001. **j.** ChIP assay was performed in SNU216 cells treated with ACTL6A shRNA or scrambled shRNA using anti-NRF2 or anti-IgG antibodies, followed by RT-qPCR with primers recognizing the predicting binding site of NRF2 in the transcriptional start of GCLC. The fold expression of ChIP-enriched mRNAs relative to the input was calculated. The data are presented as the means \pm SD, *** P <0.001.

Rebuttal Figure 14. a. Relative cell growth rate of SNU216 cells treated with ACTL6A shRNA or scrambled shRNA, and transfected with flag-ACTL6A-WT, constructs of domain deletion or pcDNA3.1 vector for the indicated time points. The data are presented as the means \pm SEM, *** P <0.001, *ns*=no significance. **b.** Immunoblotting analysis of the indicated proteins in cells from (a). **c.** Immunoblot analysis of the indicated proteins from M2 beads immunoprecipitates (IP) and whole cell lysates (input) obtained from SNU216 cells transfected with ACTL6A-WT, ACTL6A-ΔHR or pcDNA3.1 vector. **d.** ChIP assay was performed in SNU216 cells treated with ACTL6A shRNA or scrambled shRNA, and transfected with ACTL6A-WT, ACTL6A-ΔHR or pcDNA3.1 vector using anti-NRF2 or anti-IgG antibodies, followed by RT-qPCR with primers recognizing the predicting binding site of NRF2 in the transcriptional start of GCLC. The fold expression of ChIP-enriched mRNAs relative to the input was calculated. Data are

representative of three repeated experiments, $***P<0.001$, ns =no significance. **e-f.** Relative DCFH-DA (**e**) and C11-BODIPY (**f**) fluorescence measured by flow cytometry of SNU216 cells with indicated treatment. The data are presented as the means \pm SD, $**P<0.01$, $***P<0.001$.

3. Moreover, this would also encourage these analyses in non-GC GC cell lines to delineate if this mechanism is widespread or lineage-specific.
 →We appreciate the reviewer's suggestions. About non-GC cell lines, we found that ACTL6A KD could inhibit cell proliferation in two CRC cell lines (HCT116 and DLD1), whilst NAC and Fer-1 could not rescue it (Rebuttal Fig. 15a-b, 15d-e). And ACTL6A KD had no impact on GCLC mRNA expression in HCT116 and DLD1 (Rebuttal Fig. 15c, f). Therefore, we assumed that this mechanism is lineage-specific.

Rebuttal Figure 15. a. Relative cell growth rate of HCT116 cells treated with ACTL6A shRNA or scrambled shRNA and cultured with or without 100 μ M NAC for the indicated time points. The data are presented as the means \pm SEM, $***P<0.001$, ns =no significance. **b.** Relative cell growth rate of HCT116 cells treated with ACTL6A shRNA or scrambled shRNA and cultured with or without 1 μ M Fer-1 for the indicated time points. The data are presented as the means \pm SEM, $***P<0.001$, ns =no significance. **c.** mRNA expression levels of ACTL6A and GCLC in HCT116 cells treated with ACTL6A shRNA or scrambled shRNA. Data are representative of three repeated experiments, $***P<0.001$, ns =no significance. **d.** Relative cell growth rate of DLD1 cells treated with ACTL6A shRNA or scrambled shRNA and cultured with or without 100 μ M NAC for the indicated time points. The data are presented as the means \pm SEM, $***P<0.001$, ns =no significance. **e.** Relative cell growth rate of DLD1 cells treated with ACTL6A shRNA or scrambled shRNA and cultured with or without 1 μ M Fer-1 for the indicated time points. The data are presented as the means \pm SEM, $***P<0.001$, ns =no significance. **f.** mRNA

expression levels of ACTL6A and GCLC in DLD1 cells treated with ACTL6A shRNA or scrambled shRNA. Data are representative of three repeated experiments, *** $P < 0.001$, *ns*=no significance.

4. ACTL6A is reported as a common essential gene in multiple genome-wide CRISPR screens (www.depmap.org) and yet little or no co-essentiality is shared with known regulators of ferroptosis. Therefore, it's not clear if the cell death or growth impairment observed is only due to ferroptosis. Can the complete loss of ACTL6A (knockout) be rescued by liproxstatin treatment or GCLC overexpression?

→We appreciate the reviewer's comments. When we explored whether ACTL6A KD would cause other types of cell death, we found that cells ZVAD-FMK (apoptosis inhibitor) or necrostatin-1 (necroptosis inhibitor) could not rescue cell proliferation, and ACTL6A suppression did not sensitize GC cells to compounds (e.g., etoposide and doxorubicin) that inhibit cell growth through other mechanisms (Fig. 3a-f). Therefore, we believed that the cell death or growth impairment might be due to other mechanism, but ferroptosis played a more essential role in this process.

Otherwise, we built up ACTL6A knockout (KO) cells by CRISPR-cas9 gene editing. We found that ACTL6A KO inhibited cell growth significantly, which could be rescued by liproxstatin-1 treatment (Rebuttal Fig. 16a) and GCLC overexpression (Rebuttal Fig. 16b).

Rebuttal Figure 16. a. Relative cell growth rate of parental SNU638 cells and ACTL6A knockout SNU638 cells, and treated with or without liproxstatin-1 for the indicated time points.

The data are presented as the means \pm SEM, *** P <0.001. **b.** Relative cell growth rate of parental SNU638 cells and ACTL6A knockout SNU638 cells, and transfected with or without flag-GCLC for the indicated time points. The data are presented as the means \pm SEM, *** P <0.001.

5. The authors should also better characterize the ACTL6A cell in regard to other ferroptosis regulators, including GPX4, FSP1, DHODH, SLC7A11, and ACSL4 - for example.

→We appreciate the reviewer's comments. We detected GPX4, FSP1, DHODH, SLC7A11, and ACSL4 mRNA levels, which showed that ACTL6A KD inhibited GPX4 and SLC7A11 expression, but had no impact on FSP1, DHODH and ACSL4 (Rebuttal Fig. 17). From this result, we believed that ACTL6A might inhibits ferroptosis via SLC7A11 or GPX4, but GCLC was the most significant regulated gene.

Rebuttal Figure 17. mRNA expression levels of GCLC, SLC7A11, GPX4, DHODH, ACSL4 and FSP1 in GC cells treated with ACTL6A shRNA or scrambled shRNA.

6. Along similar lines, challenging ACTL6A with ferroptosis-inducing agents that inhibit GPX4, FSP1 or DHODH could be informative.

→We appreciate the reviewer's comments. Taking the reviewer's suggestions, we challenged ACTL6A with some ferroptosis-inducing agents and observed that ACTL6A suppression sensitized GC cells to RSL3 (a GPX4 inhibitor), iFSP1 (a FSP1 inhibitor) and brequinar (a DHODH inhibitor) (Rebuttal Fig. 18a-c).

Rebuttal Figure 18. a-c. Viability of SNU638 cells treated with ACTL6A shRNA or scrambled shRNA after 24 hours cultured with or without varying concentrations of RSL3 (a), iFSP1 (b) and brequinar (c). The data are presented as the means \pm SD.

Minor comments:

1. In Figure 3h, it's interesting to note that the organoids have remarkable morphological differences, especially the Fer-1 treated ones. Maybe the author care to comment on this.

→We appreciate the reviewer's comments. When ACTL6A was knocked down, organoids had a compact morphology with no lumen, and NAC or Fer-1 treatment enabled organoids regained a cystic structure. We have discussed it in the result. (Line143-144, Line177, Line201-202)

2. In Figure 6c it would encourage the authors to write the exact deletions
 →We appreciate the reviewer's suggestions. We noticed that it would be clearer if we show the exact deleted site in the figure. Since it seems hard to read if they are showed in Fig. 6c, we have showed them in Fig. 6a.

3. In Figure 6e it appears that the Flag and NRF2 blots are swapped.
 → We appreciate the reviewer's comments. We are sorry that we labeled "Flag" and "NRF2" at a reverse place. We have corrected it in Fig. 6e.

Reviewer #3 (Comments to the Author):

In the manuscript entitled “ACTL6A Protects Gastric Cancer Cells against Ferroptosis through Induction of Glutathione Synthesis”, the authors applied a series of studies to investigate the correlation between ACTL6A and ferroptosis by regulating GSH metabolism; They find ferroptosis was inhibited by ACTL6A by increasing GSH level, highlights the importance of ACTL6A in gastric cancer cell metabolism and tumorigenesis. The conclusion is potentially interesting, I have several concerns that limited my passion.

Major comments:

1. The authors make conclusions that ACTL6A regulated gastric cancer tumorigenesis, however, no GC tumorigenesis data were shown, and ACTL6A null mouse-based spontaneous tumorigenesis experiments should be applied. Xenograft of cell line/PDOs in nude mice is not sufficient to support the authors' conclusion.

→ We appreciate the reviewer's comments. GC cell lines, xenografts, PDXs, PDOs, and clinical samples (TMA) were most widely used in GC researches⁵⁻⁸. Since deletion of ACTL6A resulted in rapid lethality in mice⁹, we can't use total knockout mice for our study. Mice models for gastric cancer are limited. Until recently some gastric specific-cre mice was developed for gastric cancer^{8,10}. We do agree that ACTL6A null mouse-based spontaneous tumorigenesis experiments can enhance our findings, and the lack of KO mice experiments is a shortage of this project. Due to the spontaneous GC tumorigenesis experiments were too onerous, we used cell lines, xenografts, PDXs, PDOs, and clinical samples in our research. We will try our best to cover this shortage in the future. We have replaced “tumorigenesis” with “tumor progress”.

2. The size of the GC PDOs seems too small, only ranging from 20 μm to

80 μm , which is much smaller than published GC PDOs data. Had the authors applied further experiments to identify whether they are truly cancer PDOs?

→ We appreciate the reviewer's comments. Taking the sizes of GC PDOs in the published papers as reference, some of them were under 100 μm ^{11,12}, and some of them exceeded 100 μm ¹³. Therefore, we think that the sizes of GC PDOs differed from case to case. To further identify whether they are truly cancer PDOs, we detected CEA levels of PDOs and primary tumors by IHC staining. And we found that PDOs showed similar expression of CEA with primary tumors (**Rebuttal Fig. 19**), which indicated that they were indeed cancer PDOs.

Rebuttal Figure 19. IHC staining for CEA in primary cancer and organoids of 3 cases. Scale bars represent 50 μm .

3. The quantification and statistical methods of the PDOs should be detailed in the methods, for example, what does each dot in statistical plots indicated? One single PDO, one repeated well, or one PDO case? Moreover, all experiments must be repeated at least 3 times.

→ We appreciate the reviewer's comments. Each dot in statistical plots indicated one single PDO. After treatment with shRNA lentivirus and NAC or Fer-1, we chose 8 PDOs at similar sizes to observe. These organoids were pictured every 3 days, and the sizes were measured with Image J. All experiments were repeated in 3 cases (Rebuttal Fig. 20a-c). And we have stated more details in the Methods (Line 642-648).

Rebuttal Figure 20. a-c. Representative pictures of GC patient-derived organoid (PDO). Each experiment was repeated in 3 cases. Scale bars represent 100µm.

4. In methods, the authors claimed basic PDO culture medium already contains 1mM NAC, however, data in Fig2h-I showed a dramatic difference when adding 100 µM additional NAC, the accuracy of the data is questionable.

→ We appreciate the reviewer's comments. Actually, we added 1mM NAC when building up organoids, and then we extracted original medium and used

medium without NAC in the latter experiments. We are sorry that we did not state it clearly, and we have stated the details in the Methods (Line 642-648).

5. SNU638 cell is the only gastric cancer cell line used in this paper. Making conclusions in only one cell line is dangerous.

→ We appreciate the reviewer's comments. In our primary submission, we showed that ACTL6A promote GC cell proliferation in three cell lines (SNU638, SNU668 and SNU216) (Fig 1e-f, Supplementary Fig. 1b-c), and ACTL6A KD decreased GSH/GSSG ratio in SNU216 (Supplementary Fig. 2f). Besides, we also showed that NAC rescued ACTL6A-KD cell proliferation in SNU216 (Supplementary Fig. 2g). And in Supplementary Fig. 3a, we showed that Fer-1 rescued ACTL6A-KD cell viability in SNU216.

To further validate our findings, we conducted more experiments in another GC cell line (SNU216). First of all, ACTL6A KD increased cellular ROS levels and lipid peroxidation levels of SNU216 (Rebuttal Fig. 12a-b). ACTL6A regulates GCLC in transcriptional level via NRF2 (Rebuttal Fig. 13a-j). The HR domain of ACTL6A is essential in GCLC regulating and ferroptosis (Rebuttal Fig. 14a-f). The results indicated that our findings could be repeated in another cell line. Please refer to reviewer2 major question2.

6. The ACTL6A seems nuclear expression in Fig 3i but cell plasma expression in Fig 7q. The authors should explain the deviation between them.

→ We appreciate the reviewer's comments. The ACTL6A protein mostly localizes to nuclear, and some of ACTL6A localizes to cytosol. Due to tumor heterogeneity, a small quantity of GC tissues in our TMA showed that ACTL6A seems cell plasma expression. We mistakenly chose to show the atypical pictures. We have showed pictures that ACTL6A was typical expression: most nuclear and some cytosol localization (Rebuttal Fig. 21). And we have replaced the pictures in Fig 7q.

Rebuttal Figure 21. Representative images of GC TMA that IHC stained with ACTL6A and GCLC. Scale bars represent 50 μ m.

Minor comments:

1. Measured the grayscales in all Western bolt bands.

→ We appreciate the reviewer's comments. We have measured the grayscales of all western bolt bands and showed them in figures.

2. X axis in Fig 1d is missed.

→ We appreciate the reviewer's comments. In Fig 1d, each column represents one case of GC patients. We have added X axis in Fig 1d.

REFERENCES

- 1 Comprehensive molecular characterization of gastric adenocarcinoma. *Nature* **513**, 202–209, doi:10.1038/nature13480 (2014).
- 2 Liu, Y. *et al.* Identification of ferroptosis as a novel mechanism for antitumor activity of natural product derivative a2 in gastric cancer. *Acta pharmaceutica Sinica. B* **11**, 1513–1525, doi:10.1016/j.apsb.2021.05.006 (2021).

- 3 Ogiwara, H. *et al.* Targeting the Vulnerability of Glutathione
Metabolism in ARID1A-Deficient Cancers. *Cancer cell* **35**,
177–190. e178, doi:10.1016/j.ccell.2018.12.009 (2019).
- 4 Sayin, V. I. *et al.* Antioxidants accelerate lung cancer progression
in mice. *Science translational medicine* **6**, 221ra215,
doi:10.1126/scitranslmed.3007653 (2014).
- 5 Shi, W. *et al.* Hyperactivation of HER2–SHCBP1–PLK1 axis promotes
tumor cell mitosis and impairs trastuzumab sensitivity to gastric
cancer. *Nature communications* **12**, 2812,
doi:10.1038/s41467-021-23053-8 (2021).
- 6 Wang, Q. *et al.* METTL3-mediated m(6)A modification of HDGF mRNA
promotes gastric cancer progression and has prognostic
significance. *Gut* **69**, 1193–1205, doi:10.1136/gutjnl-2019-319639
(2020).
- 7 Cao, T. *et al.* A CGA/EGFR/GATA2 positive feedback circuit confers
chemoresistance in gastric cancer. *The Journal of clinical
investigation* **132**, doi:10.1172/jci154074 (2022).
- 8 Liabeuf, D., Oshima, M., Stange, D. E. & Sigal, M. Stem Cells,
Helicobacter pylori, and Mutational Landscape: Utility of
Preclinical Models to Understand Carcinogenesis and to Direct
Management of Gastric Cancer. *Gastroenterology* **162**, 1067–1087,
doi:10.1053/j.gastro.2021.12.252 (2022).
- 9 Krasteva, V. *et al.* The BAF53a subunit of SWI/SNF-like BAF complexes
is essential for hemopoietic stem cell function. *Blood* **120**,
4720–4732, doi:10.1182/blood-2012-04-427047 (2012).
- 10 Fatehullah, A. *et al.* A tumour-resident Lgr5(+) stem-cell-like pool
drives the establishment and progression of advanced gastric
cancers. *Nat Cell Biol* **23**, 1299–1313,
doi:10.1038/s41556-021-00793-9 (2021).
- 11 Seidlitz, T. *et al.* Human gastric cancer modelling using organoids.
Gut **68**, 207–217, doi:10.1136/gutjnl-2017-314549 (2019).
- 12 Zou, J. *et al.* Construction of gastric cancer patient-derived
organoids and their utilization in a comparative study of
clinically used paclitaxel nanoformulations. *Journal of
nanobiotechnology* **20**, 233, doi:10.1186/s12951-022-01431-8 (2022).
- 13 Song, H. *et al.* Establishment of Patient-Derived Gastric Cancer
Organoid Model From Tissue Obtained by Endoscopic Biopsies. *Journal
of Korean medical science* **37**, e220, doi:10.3346/jkms.2022.37.e220
(2022).

REVIEWER COMMENTS

Reviewer #1 (Remarks to the Author):

The manuscript was improved by additional experimental and this reviewer commends the efforts by the authors. However, there are a couple of comments remain to be addressed satisfactorily.

1. The criteria for defining high vs. low ACTL6A expression used by the authors is not practical and inconsistent with the rest of the manuscript. The authors stated in the rebuttal that high is defined as >2-fold over adjacent normal, yet the rest of analysis are among a group of tumors. For the conclusion from the manuscript to be practical, a defined criteria to differentiate high vs. low ACTL6A is essential. Otherwise, it's hard to believe the marginal difference in ACTL6A expression observed in Fig. 7a can account for the differences observed in response to BSO.

2. Please perform a genome-wide analysis for pathways commonly regulated by ACTL6A and BRG1 based on the newly obtained BRG1 ChIP-seq analysis. It's informative to determine what other pathways regulated by both ACTL6A and BRG1.

Reviewer #2 (Remarks to the Author):

The authors have satisfactorily addressed most of my points. I have nothing urgent that requires to be addresses at this stage.

Reviewer #4 (Replacement reviewer to comment on behalf of Reviewer #3, Remarks to the Author):

The manuscript 'ACTL6A Protects Gastric Cancer Cells against Ferroptosis through Induction of Glutathione Synthesis' by Yang et al. demonstrates that an oncogene ACTL6A reduces ROS levels by regulating the metabolism of GSH, an antioxidant, specifically by upregulating transcription of GCLC as a co-transcriptional factor along with NRF2, which can promote cancer survival and prevent ROS-induced ferroptosis. This study expands our understanding of ROS pathways for which established therapeutics, such as NAC and BSO, exist. The current version of the data does not contain any obvious errors. Several

concerns that need be addressed have more to do with the fact that the authors are focusing on gastric cancer than any other type of cancer.

1) Tumor characteristics or patient demographics used in the study are missing, limiting the relevance of this study for gastric cancer. What was the Lauren classification or molecular subtype of molecular tumor for gastric tumors used in the study? Are the samples from the antrum or the corpus? Is there a significant association between ACTL6A expression and Helicobacter pylori infection status, atrophy, or metaplasias such as intestinal or pseudopyloric metaplasia of the stomach?

2) Page 3, "ACTL6A, which encodes a SWI/SNF component associated with stem cell and progenitor cell activities, was substantially expressed in primary GC patient specimens": Does ACTL6A just influence stem and progenitor populations and not differentiated cells? What is the expression pattern of ACTL6A in the stomach at homeostasis in the corpus and antrum?

3) Figure 1f: how does overexpression of ACTL6A promote cell proliferation? Does reducing ROS levels below the usual level promote proliferation? Does it occur through a different mechanism? Please elaborate.

Reviewer #1 (Remarks to the Author)

The manuscript was improved by additional experimental and this reviewer commends the efforts by the authors. However, there are a couple of comments remain to be addressed satisfactorily.

1. The criteria for defining high vs. low ACTL6A expression used by the authors is not practical and inconsistent with the rest of the manuscript. The authors stated in the rebuttal that high is defined as >2-fold over adjacent normal, yet the rest of analysis are among a group of tumors. For the conclusion from the manuscript to be practical, a defined criteria to differentiate high vs. low ACTL6A is essential. Otherwise, it's hard to believe the marginal difference in ACTL6A expression observed in Fig. 7a can account for the differences observed in response to BS0.

→A: We appreciate the reviewer's comments. To define high vs. low ACTL6A expression in PDX experiments, we detected ACTL6A protein expression of 48 cases GC samples, and the quantification is shown (Response Fig. 1). The data passed the normal distribution test. The data higher than the upper quartile are defined as high ACTL6A group, and those lower than the lower quartile are defined as low ACTL6A group (normalized to the median). In our PDX experiment, the ACTL6A expression of case 1 and case 3 are among the high ACTL6A group, and case 2 and case 4 are among the low ACTL6A group. And we have supplemented it in the Methods (Line 631-635).

Response Figure 1. Immunoblot analysis of ACTL6A protein levels in 48 GC samples. The quantification (normalized to the median) is shown.

2. Please perform a genome-wide analysis for pathways commonly regulated by ACTL6A and BRG1 based on the newly obtained BRG1 ChIP-seq analysis. It's informative to determine what other pathways regulated by both ACTL6A and BRG1.

→A: We appreciate the reviewer's comments. To determine what pathways are commonly regulated by both ACTL6A and BRG1, we overlapped the genes with loci peak in anti-ACTL6A ChIP-seq and anti-BRG1 ChIP-seq results (Response Fig.2a). And then we did an enrichment analysis of the 12209 overlapped genes. We showed the top 10 enriched KEGG pathways commonly regulated by both ACTL6A and BRG1, including: 1. Endocytosis; 2. Cell cycle; 3. Ubiquitin mediated proteolysis; 4. Hippo signaling pathway; 5. Proteoglycans in cancer; 6. Small cell lung cancer; 7. Axon guidance; 8.

MAPK signaling pathway; 9. Hepatocellular carcinoma; 10. Focal adhesion (Response Fig.2b). And top 10 enriched GO pathways in 3 aspects (Biological process, Cellular component and Molecular function) were shown in Response Fig.2c.

Response Figure 2. a-c. Venn diagram illustrating the overlapped genes in anti-ACTL6A ChIP-seq and anti-BRG1 ChIP-seq results (**a**). Top 10 enriched KEGG pathways (**b**) and GO pathways in 3 aspects (Biological process, Cellular component and Molecular function) (**c**) of 12209 overlapped genes analyzed by the package "cluster profile" of R programming language. False discovery rate (FDR)<0.05, q<0.001.

Reviewer #4 (Replacement reviewer to comment on behalf of Reviewer #3, Remarks to the Author):

The manuscript 'ACTL6A Protects Gastric Cancer Cells against Ferroptosis through Induction of Glutathione Synthesis' by Yang et al. demonstrates that an oncogene ACTL6A reduces ROS levels by regulating the metabolism of GSH, an antioxidant, specifically by upregulating transcription of GCLC as a co-transcriptional factor along with NRF2, which can promote cancer survival and prevent ROS-induced ferroptosis. This study expands our understanding of ROS pathways for which established therapeutics, such as NAC and BSO, exist. The current version of the data does not contain any obvious errors. Several concerns that need be addressed have more to do with the fact that the authors are focusing on gastric cancer than any other type of cancer.

1) Tumor characteristics or patient demographics used in the study are missing, limiting the relevance of this study for gastric cancer. What was the Lauren classification or molecular subtype of molecular tumor for gastric tumors used in the study? Are the samples from the antrum or the corpus? Is there a significant association between ACTL6A expression and Helicobacter pylori infection status, atrophy, or metaplasias such as intestinal or pseudopyloric metaplasia of the stomach?

→A: Thanks a lot for the reviewer's suggestions. We collected the WHO pathological classifications of GC samples used in the study, and found that most of them (77.7%, 143/184) were adenocarcinomas (including papillary adenocarcinomas, tubular adenocarcinomas and mucinous adenocarcinomas), 15.8% (29/184) of them were signet-ring cell carcinomas, the rest of them (6.5%, 12/184) were adenosquamous carcinomas, squamous

cell carcinomas, undifferentiated carcinomas, small cell neuroendocrine carcinomas, etc. As for the tumor location, about half of the samples (48.4%, 89/184) were from the antrum, 26.1% (48/184) were from the corpus, 21.7% (40/184) were from the fundus, and 3.8% (7/184) were from the whole stomach or multiple areas. There is no significant association between ACTL6A expression and pathological classifications or tumor location. And the information above has been added to the Supplementary Table 4.

Supplementary Table 4. Correlation between expression of ACTL6A-GCLC axis and clinicopathological features of gastric cancer patients.

Variable	ACTL6A expression		GCLC expression	
	Low	High p value ^a	Low	High p value ^a
Gender		0.676		0.402
Male	70	59	65	64
Female	28	27	24	31
age		0.736		0.360
<59 years	33	31	28	36
≥59 years	65	55	61	59
Histological grade		0.087		0.128
G1	0	5	2	3
G2	19	12	21	10
G3	72	62	60	74
G4	7	7	6	8
pT status		0.557		0.755
T1	3	1	3	1
T2	7	8	8	7
T3	55	54	52	57
T4	33	23	26	30
pN status		0.562		0.296

N0	27	17	26	18
N1	14	17	14	17
N2	21	20	21	20
N3	36	32	28	40
pM status		0.417		0.324
M0	92	78	84	86
M1	6	8	5	9
Clinical stage		0.451		0.578
I	6	2	5	3
II	30	31	32	29
III	56	45	47	54
IV	6	8	5	9
Tumor location		0.977		0.578
Antrum	48	41	41	48
Corpus	26	22	24	24
Fundus	20	20	22	18
Whole stomach	4	3	2	5
Pathological type		0.613		0.106
Adenocarcinoma ^b	78	65	75	68
SRC carcinoma ^c	13	16	9	20
Others ^d	7	5	5	7

NOTE: All data are no. of patients.

^a*p* values were calculated in SPSS16.0 using a Pearson Chi-Square Test (Fisher's Exact Test was used when >20% cells have expected count less than 5). *p* values <0.05 were considered to indicate statistical significance.

^b Adenocarcinoma includes papillary adenocarcinoma, tubular adenocarcinoma and mucinous adenocarcinoma.

^c SRC carcinoma, signet-ring cell carcinoma.

^d Others pathological classifications includes adenosquamous carcinoma, squamous cell carcinoma, undifferentiated carcinoma, small cell neuroendocrine carcinoma, etc.

The information about *Helicobacter pylori* infection status, atrophy, and metaplasias were not included when we collected the clinicopathological features. Then we searched the GEO datasets, and found that in GSE129219, there is no significant association between ACTL6A expression and *Helicobacter pylori* infection status (Response Fig.3).

Response Figure 3. Gene expression of ACTL6A in GC samples with or without *H. pylori* infection from GSE129219. FPKM, Fragments Per Kilobase of exon model per Million mapped fragments. *ns*=no significance.

2) Page 3, "ACTL6A, which encodes a SWI/SNF component associated with stem cell and progenitor cell activities, was substantially expressed in primary GC patient specimens": Does ACTL6A just influence stem and progenitor populations and not differentiated cells? What is the expression pattern of ACTL6A in the stomach at homeostasis in the corpus and antrum?

→A: We appreciate the reviewer's comments. It was previously reported that ACTL6A is essential for hemopoietic stem cell function and neural progenitor proliferation^{1,2}. And our previous study showed that ACTL6A protects embryonic stem cells from differentiating into primitive endoderm³. There is little research about the function of ACTL6A in differentiated cells of non-cancer diseases. However, in recent years, ACTL6A has been reported to be related to the tumorigenesis of several cancers⁴⁻⁹. Since cancer cells display

various differentiation status, ACTL6A might also influence differentiated cells.

In GEO dataset GSE141657, gene expression in three biological replicates of six different conditions (undifferentiated and differentiated antrum, corpus and fundus samples) was determined by RNA array. We found that ACTL6A expressed a little higher in undifferentiated antrum than in differentiated antrum. It seemed no significant association between the differentiation state and the expression of ACTL6A in gastric corpus and fundus samples

(Response Fig.4)¹⁰.

Response Figure 4. Gene expression of ACTL6A in undifferentiated and differentiated gastric antrum, corpus and fundus samples from GSE141657. Each sample contains 3 replicates.

3) Figure 1f: how does overexpression of ACTL6A promote cell proliferation? Does reducing ROS levels below the usual level promote proliferation? Does it occur through a different mechanism? Please elaborate.

→A: We appreciate the reviewer's comments. An increase in ROS is associated with abnormal cancer cell growth. If the increase of ROS reaches a certain threshold level that is incompatible with cellular survival, ROS may

exert a cytotoxic effect, leading to the death of malignant cells and thus limiting cancer progression¹¹. Significantly high ROS production occur in gastric cancer cells due to *H. pylori* infections, EBV infections or other mechanisms¹². Therefore, GC cells exhibited high ROS levels, and overexpression of ACTL6A could reduce it and prevent ferroptosis, then promote proliferation. Since some cancer cells might become well-adapted under persistent intrinsic oxidative stress¹¹, overexpression of ACTL6A might also promote proliferation through a different mechanism. It deserves more investigation. We have discussed it in the manuscript (Line 411-415).

References

- 1 Krasteva, V. *et al.* The BAF53a subunit of SWI/SNF-like BAF complexes is essential for hemopoietic stem cell function. *Blood* **120**, 4720-4732, doi:10.1182/blood-2012-04-427047 (2012).
- 2 Lessard, J. *et al.* An essential switch in subunit composition of a chromatin remodeling complex during neural development. *Neuron* **55**, 201-215, doi:10.1016/j.neuron.2007.06.019 (2007).
- 3 Lu, W. *et al.* Actl6a protects embryonic stem cells from differentiating into primitive endoderm. *Stem cells (Dayton, Ohio)* **33**, 1782-1793, doi:10.1002/stem.2000 (2015).
- 4 Meng, L. *et al.* BAF53a is a potential prognostic biomarker and promotes invasion and epithelial-mesenchymal transition of glioma cells. *Oncology reports* **38**, 3327-3334, doi:10.3892/or.2017.6019 (2017).
- 5 Saladi, S. V. *et al.* ACTL6A Is Co-Amplified with p63 in Squamous Cell Carcinoma to Drive YAP Activation, Regenerative Proliferation, and Poor Prognosis. *Cancer cell* **31**, 35-49, doi:10.1016/j.ccell.2016.12.001 (2017).
- 6 Zeng, Z., Yang, H. & Xiao, S. ACTL6A expression promotes invasion, metastasis and epithelial mesenchymal transition of colon cancer. *BMC cancer* **18**, 1020, doi:10.1186/s12885-018-4931-3 (2018).
- 7 Xiao, S. *et al.* Actin-like 6A predicts poor prognosis of hepatocellular carcinoma and promotes metastasis and epithelial-mesenchymal transition. *Hepatology (Baltimore, Md.)* **63**, 1256-1271, doi:10.1002/hep.28417 (2016).
- 8 Shrestha, S., Adhikary, G., Xu, W., Kandasamy, S. & Eckert, R. L. ACTL6A suppresses p21(Cip1) expression to enhance the epidermal squamous cell carcinoma phenotype. *Oncogene*, doi:10.1038/s41388-020-1371-8 (2020).
- 9 Park, J., Wood, M. A. & Cole, M. D. BAF53 forms distinct nuclear complexes and functions as a critical c-Myc-interacting nuclear cofactor for oncogenic transformation. *Molecular and cellular biology* **22**, 1307-1316, doi:10.1128/mcb.22.5.1307-1316.2002 (2002).

- 10 Fritsche, K. *et al.* DNA methylation in human gastric epithelial cells defines regional identity without restricting lineage plasticity. *Clinical epigenetics* **14**, 193, doi:10.1186/s13148-022-01406-4 (2022).
- 11 Trachootham, D., Alexandre, J. & Huang, P. Targeting cancer cells by ROS-mediated mechanisms: a radical therapeutic approach? *Nature reviews. Drug discovery* **8**, 579-591, doi:10.1038/nrd2803 (2009).
- 12 Bhattacharyya, A., Chattopadhyay, R., Mitra, S. & Crowe, S. E. Oxidative stress: an essential factor in the pathogenesis of gastrointestinal mucosal diseases. *Physiological reviews* **94**, 329-354, doi:10.1152/physrev.00040.2012 (2014).

REVIEWERS' COMMENTS

Reviewer #1 (Remarks to the Author):

This reviewer thanks the reviewer for their efforts. However, this reviewer remains skeptical with regard to the authors' definition of high vs. low ACTL6A, the foundation of this manuscript. In the new rebuttal data, the authors yet used a completely different criteria to define high vs. low (top vs bottom quartile). Regardless, the differences in high vs low is marginal in immunoblot. This might not account for the observed difference in pathways/treatment etc. Instead, the new data would suggest other pathways/genes might account for the differences.

Reviewer #4 (Remarks to the Author):

The authors have addressed the issues raised.

Reviewer #1 (Remarks to the Author)

This reviewer thanks the reviewer for their efforts. However, this reviewer remains skeptical with regard to the authors' definition of high vs. low ACTL6A, the foundation of this manuscript. In the new rebuttal data, the authors yet used a completely different criteria to define high vs. low (top vs bottom quartile). Regardless, the differences in high vs low is marginal in immunoblot. This might not account for the observed difference in pathways/treatment etc. Instead, the new data would suggest other pathways/genes might account for the differences.

→A: We appreciate the reviewer's comments. In a previous study¹, they made definition criteria of high vs. low target molecule (MAP3K8) in PDX. MEK inhibitors markedly reduced tumor growth in high-MAP3K8 PDX models (Response Fig 1a-d). Besides, our previous study showed that administration of the SGOC pathway inhibitor NCT-503 in the established ILF3-high PDX tumors attenuated tumor progression (Response Fig 2a-b)². Similarly, in this study, the expression levels of ACTL6A in PDX case 1 and case 3 (ACTL6A high) are over 4 times as those in PDX case 2 and case 4 (ACTL6A low). GCLC inhibitors (BSO) treatment inhibits tumor growth of ACTL6A-high PDXs. And we showed that the ACTL6A expression levels of PDX cases meets both of the definition criteria (top vs bottom quartile and T/N >2 times fold change as high).

Indeed, there might be other pathways/genes account for the differences. For example, pathways involved in glutathione metabolism or GCLC regulation might also be modified in the cases, which would affect the sensitivity of PDX tumors to BSO. Besides, other pathways/genes that influence redox balance or ferroptosis might also account for the differences. Based on the function of ACTL6A on GCLC regulating, we believe that the expression of ACTL6A plays an important role in it.

We have discussed it in the Discussion section (Line 403-414).

Response Figure 1. Figures from a cited paper¹. **a.** Western blots showing MAP3K8, P-MEK and MEK protein levels in seven different mouse models of patient-derived xenograft (PDX). **b.** Scatter plots of MAP3K8 protein levels in tumors from the seven PDX mouse models studied. **c-d.** Relative tumor volume over time of PDX models exhibiting either high- (PDX1) (**c**) or low- (PDX2) (**d**) MAP3K8 protein levels.

Response Figure 2. Figures from our previous study². **a.** Expression level of ILF3 in indicated patient-derived xenografts (PDXs). **b.** Impact of NCT-503 on tumor growth in mice (n = 7/group) bearing indicated PDXs.

References:

- 1 Guosso, T. *et al.* MAP3K8/TPL-2/COT is a potential predictive marker for MEK inhibitor treatment in high-grade serous ovarian carcinomas. *Nature communications* **6**, 8583, doi:10.1038/ncomms9583 (2015).
- 2 Li, K. *et al.* ILF3 is a substrate of SPOP for regulating serine biosynthesis in colorectal cancer. *Cell research* **30**, 163-178, doi:10.1038/s41422-019-0257-1 (2020).